# Variance-Aware Feel-Good Thompson Sampling for Contextual Bandits

**Xuheng Li**
Department of Computer Science
University of California, Los Angeles
California, 90095
xuheng.li@cs.ucla.edu

**Quanquan Gu**
Department of Computer Science
University of California, Los Angeles
California, 90095
qgu@cs.ucla.edu

## Abstract

Variance-dependent regret bounds have received increasing attention in recent studies on contextual bandits. However, most of these studies are focused on upper confidence bound (UCB)-based bandit algorithms, while sampling based bandit algorithms such as Thompson sampling are still understudied. The only exception is the `LinVDTS` algorithm (Xu et al., 2023), which is limited to linear reward function and its regret bound is not optimal with respect to the model dimension. In this paper, we present `FGTS-VA`, a variance-aware Thompson Sampling algorithm for contextual bandits with general reward function with optimal regret bound. At the core of our analysis is an extension of the decoupling coefficient, a technique commonly used in the analysis of Feel-good Thompson sampling (FGTS) that reflects the complexity of the model space. With the new decoupling coefficient denoted by $\mathrm{dc}$, `FGTS-VA` achieves the regret of $\widetilde{\mathcal{O}}\big(\sqrt{\mathrm{dc} \cdot \log |\mathcal{F}| \sum_{t=1}^{T} \sigma_t^2} + \mathrm{dc}\big)$, where $|\mathcal{F}|$ is the size of the model space, $T$ is the total number of rounds, and $\sigma_t^2$ is the subgaussian norm of the noise (e.g., variance when the noise is Gaussian) at round $t$. In the setting of contextual linear bandits, the regret bound of `FGTS-VA` matches that of UCB-based algorithms using weighted linear regression (Zhou and Gu, 2022).

## 1 Introduction

The contextual bandit (Langford and Zhang, 2007) is a pivotal setting in interactive decision making, and is an important generalization of the multi-armed bandit that incorporates context-dependent reward functions. However, the standard contextual bandit setting is unable to account for heterogeneous and context-dependent noise of the observed rewards, which can have significant impacts on the performance of algorithms (Auer, 2002). Additionally, algorithms designed for standard contextual bandits (Abbasi-Yadkori et al., 2011; Chu et al., 2011) are usually incompatible with potentially benign environments, despite achieving the minimax regret bound in the worst case. For example, these algorithms have a worst-case regret bound of $\widetilde{\mathcal{O}}(d\sqrt{T})$ for linear contextual bandits, where $T$ is the number of steps; however, if the reward is deterministic, then the algorithm based on simple exploration can achieve the regret of $\widetilde{\mathcal{O}}(d)$. To fill this gap, a number of approaches (Zhou et al., 2021; Zhang et al., 2021; Zhou and Gu, 2022; Kim et al., 2022; Xu et al., 2023; Zhao et al., 2023; Jia et al., 2024) have been developed to account for the heterogeneous magnitudes of the noise, and regret bounds are established depending on $\sigma_t^2$, i.e., the noise variance in step $t$. Most of these approaches are based on the upper confidence bound (UCB). Notably, Zhou and Gu (2022) and Zhao et al. (2023) established the nearly optimal regret bound of $\widetilde{\mathcal{O}}\big(d\sqrt{\sum_{t=1}^{T} \sigma_t^2} + d\big)$ in linear bandits, which degenerates to $\widetilde{\mathcal{O}}(d)$ in the deterministic case, under the settings where $\sigma_t^2$ are known and agnostic to the agent, respectively.

Thompson sampling (TS) (Thompson, 1933) is another technique that facilitates exploration of the action space apart from UCB-based methods. In TS-based algorithms, an estimation of the reward

function is sampled from the posterior distribution instead of being deterministically constructed. TS-based algorithms have displayed better efficiency than UCB-based algorithms empirically (Chapelle and Li, 2011; Osband and Van Roy, 2017), and have been extensively studied for both multi-armed bandits (Agrawal and Goyal, 2012; Kaufmann et al., 2012; Agrawal and Goyal, 2017; Jin et al., 2021) and linear bandits (Agrawal and Goyal, 2013). However, a minimax-optimal frequentist regret bound of standard TS has been lacking, and Zhang (2022) constructed an instance such that standard TS is suboptimal. To resolve this issue, Zhang (2022) proposed a new variant of TS called the **Feel-Good Thompson sampling (FGTS)**, and theoretically justifies that the frequentist regret of FGTS is $\widetilde{\mathcal{O}}(d\sqrt{T})$, which is minimax optimal and similar to UCB-based algorithms (e.g., OFUL (Abbasi-Yadkori et al., 2011) and LinUCB (Li et al., 2010)).

Despite the success the success of TS-based algorithms in the vanilla contextual bandit setting, there is a scarcity of results concerning variance-aware contextual bandits. A notable exception is Xu et al. (2023), which proposed a variant of TS with weighted ridge regression. However, this result is restricted to linear contextual bandits, and the regret is suboptimal in the model dimension $d$, which is an issue shared by TS-based algorithms designed for vanilla contextual bandits (Agrawal and Goyal, 2013; Abeille and Lazaric, 2017). Therefore, the following open question arises:

> *Is it possible to design a FGTS-based algorithm for contextual bandits whose regret is both optimal in $d$ and variance-dependent, similar to UCB-based algorithms?*

In this paper, we answer this question affirmatively with the first variance-aware algorithm based on FGTS and new analysis techniques. We summarize our contributions as follows, with a comparison of our algorithm against related algorithms shown in Table 1:

1. We propose an FGTS-based algorithm called `FGTS-VA` for the setting of variance-dependent contextual bandits, which is applicable to the general reward function class. Compared with the standard FGTS algorithm in Zhang (2022), the posterior distribution of `FGTS-VA` adopts not only variance-dependent weights applied to the log-likelihoods but also a new **feel-good exploration term**. When reduced to standard contextual bandits, `FGTS-VA` is the first FGTS-based algorithm that does not require knowledge of the horizon $T$.

2. In the analysis of `FGTS-VA`, we propose the **generalized decoupling coefficient**, which is a novel extension of the standard decoupling coefficient commonly used in the analysis of FGTS. We relate the generalized decoupling coefficient to other complexity measures by showing that (i) the generalized decoupling coefficient is $\widetilde{\mathcal{O}}(d)$ for linear contextual bandits, and that (ii) it is bounded by the generalized Eluder dimension for the general reward function class.

3. Equipped with the generalized decoupling coefficient (denoted as $\mathrm{dc}$), we show that the regret bound of `FGTS-VA` is $\mathcal{O}\big(\sqrt{(1 + \sum_{t=1}^{T} \sigma_t^2)\,\mathrm{dc}\log|\mathcal{F}|} + \mathrm{dc}\big)$ in expectation, where $|\mathcal{F}|$ is the cardinality of the function class. For linear contextual bandits, `FGTS-VA` enjoys the nearly optimal regret of $\widetilde{\mathcal{O}}\big(d\sqrt{\sum_{t=1}^{T}\sigma_t^2} + d\big)$, similar to UCB-based algorithms (Zhou and Gu, 2022; Zhao et al., 2023). When restricted to the deterministic case, the regret of `FGTS-VA` is $\widetilde{\mathcal{O}}(\mathrm{dc})$, matching the lower bound given by Xu et al. (2023).

**Notation.** We use $\mathbb{1}[\cdot]$ to denote the indicator function. We use $\mathrm{KL}(\cdot||\cdot)$ to denote the KL-divergence of two distributions. We use standard asymptotic notations $\mathcal{O}(\cdot)$, $\Omega(\cdot)$, and $\Theta(\cdot)$, with $\widetilde{\mathcal{O}}(\cdot)$, $\widetilde{\Omega}(\cdot)$, and $\widetilde{\Theta}(\cdot)$ hiding logarithmic factors; $f(\cdot) \lesssim g(\cdot)$ means $f(\cdot) = \mathcal{O}(g(\cdot))$. We use non-boldface letters to denote scalars, boldface lower-case letters to denote vectors, and boldface upper-case letters to denote matrices. We use $\langle \cdot, \cdot \rangle$ to denote the inner product, i.e., for vectors $\mathbf{a}$ and $\mathbf{b}$, define $\langle \mathbf{a}, \mathbf{b} \rangle = \mathbf{a}^\top \mathbf{b}$; for matrices $\mathbf{A}$ and $\mathbf{B}$, define $\langle \mathbf{A}, \mathbf{B} \rangle = \mathrm{tr}(\mathbf{A}\mathbf{B}^\top)$. For a vector $\mathbf{v}$ and a positive semi-definite (PSD) matrix $\mathbf{M}$, let $\|\mathbf{v}\|_{\mathbf{M}} = \sqrt{\mathbf{v}^\top \mathbf{M} \mathbf{v}}$. For a positive integer $n$, let $[n]$ denote the set of $\{1, 2, \ldots, n\}$.

## 2 Related Work

**Variance-aware algorithms.** Audibert et al. (2009) proposed the first algorithm that utilized the variance information through empirical estimates of the variance, with a line of works based on similar techniques for various settings (Hazan and Kale, 2011; Wei and Luo, 2018; Ito, 2021; Ito and Takemura, 2023). Mukherjee et al. (2018) used the variance estimates to characterize confidence intervals and to perform arm elimination.

Table 1: Comparison of variance-aware algorithms for bandits. We compare the regret under the setting of both stochastic and deterministic linear bandits, where $d$ is the model dimension, $T$ is the number of steps, and $\Lambda = \sum_{t=1}^{T} \sigma_t^2$ is the sum of variances. The last column stands for whether the variance is revealed to the learning agent at step $t$.

| Algorithm | Technique | Regret (General) | Regret (Deterministic) | $\sigma_t^2$ |
|---|---|---|---|---|
| Weighted OFUL+ 
 (Zhou and Gu, 2022) | UCB | $\widetilde{\mathcal{O}}(d\sqrt{\Lambda} + d)$ | $\widetilde{\mathcal{O}}(d)$ | Known |
| SAVE 
 (Zhao et al., 2023) | UCB | $\widetilde{\mathcal{O}}(d\sqrt{\Lambda} + d)$ | $\widetilde{\mathcal{O}}(d)$ | Unknown |
| LinVDTS 
 (Xu et al., 2023) | TS | $\widetilde{\mathcal{O}}(d^{1.5}\sqrt{\Lambda} + d^{1.5})$ | $\widetilde{\mathcal{O}}(d^{1.5})$ | Unknown |
| FGTS 
 (Zhang, 2022) | TS | $\widetilde{\mathcal{O}}(d\sqrt{T})$ | $\widetilde{\mathcal{O}}(d\sqrt{T})$ | NA |
| FGTS-VA 
 (This work) | TS | $\widetilde{\mathcal{O}}(d\sqrt{\Lambda} + d)$ | $\widetilde{\mathcal{O}}(d)$ | Known |

A recent line of works study variance-dependent algorithms for bandits with function approximation. For the case of known variances, a line of works have utilized the variances in weighted ridge regression in linear bandits (Zhou et al., 2021; Zhou and Gu, 2022). For the case of unknown variances, Zhang et al. (2021) and Kim et al. (2022) constructed variance-dependent confidence sets, and Zhao et al. (2023) designed a SupLin-type algorithm and proposed the idea of classifying samples into different variance levels. Di et al. (2023) used the idea of Zhao et al. (2023) to develop a variance-aware algorithm for dueling bandits. Notably, Zhou and Gu (2022) and Zhao et al. (2023) managed to derive the nearly optimal regret bounds of $\widetilde{\mathcal{O}}\big(d\sqrt{\sum_{t=1}^{T}\sigma_t^2} + d\big)$ for the two cases, respectively. For contextual bandits with the general function class, Wei et al. (2020) focused on the case where the action space is small and derived the regret bound related to the total estimation error. A recent work (Jia et al., 2024) studied two cases that are referred to the weak adversary and the strong adversary, depending on whether the actions of the agent can affect variances. This work managed to establish a regret bound of $\widetilde{\mathcal{O}}\big(d_{\text{elu}}\sqrt{\sum_{t=1}^{T}\sigma_t^2} + d_{\text{elu}}\big)$ for the strong adversary, where $d_{\text{elu}}$ stands for the Eluder dimension. However, the analysis of the weak adversary setting, which is more closely related to the setting of our work, is restricted to the case of finite action space.

Another line of works derived second-order bounds for Markov decision processes. Wang et al. (2024a) derived the first second-order bound for distributional RL. Built on the pivotal triangle inequality in Wang et al. (2024a), Wang et al. (2024b) proved that OMLE algorithm (Liu et al., 2023) without weighted regression enjoys the second-order regret bound.

**Feel-Good Thompson sampling (FGTS).** Zhang et al. (2021) first proposed FGTS, and achieved the minimax-optimal regret bound for linear contextual bandits by virtue of the feel-good exploration term in the posterior distribution. Fan and Gu (2023) extended the technique to function approximation of the policy space and applied the algorithm to a number of variants of linear contextual bandits. A recent work by Li et al. (2024) derived an algorithm for contextual dueling bandits based on FGTS that is efficient both theoretically and empirically. By replacing the feel-good exploration term with the value function in the first step, Agarwal and Zhang (2022); Dann et al. (2021) applied FGTS to model-based RL and model-free RL, respectively, with the technique of Agarwal and Zhang (2022) more similar to Zhang (2022), and the technique of Dann et al. (2021) motivating of our algorithm.

**Variance-aware Thompson sampling algorithms.** Saha and Kveton (2023) proposed a variance-aware TS-based algorithm for multi-armed bandits. Xu et al. (2023) proposed LinVDTS, which is the only algorithm for linear bandits based on Thompson sampling. The posterior distribution in LinVDTS is Gaussian with the mean estimated with the weighted ridge regression in Zhou and Gu (2022). However, the regret bound of LinVDTS is $\widetilde{\mathcal{O}}(d^{1.5}\sqrt{\sum_{t=1}^{T}\sigma_t^2} + d^{1.5})$, which is suboptimal in $d$, similar to the regret bound of $\widetilde{\mathcal{O}}(d^{1.5}\sqrt{T})$ of standard TS algorithms for linear bandits (Agrawal and Goyal, 2013; Abeille and Lazaric, 2017).

## 3 Preliminaries

### 3.1 Contextual Bandits

We study the setting of contextual bandits (Langford and Zhang, 2007) which is an extension to multi-armed bandits by adding the notion of the context set $\mathcal{X}$ and allowing the action set to be a context-dependent subset of the whole action set $\mathcal{A}$. In step $t$ over a total of $T$ steps, the agent first receives the context $x_t$ and the action set $\mathcal{A}_t \subset \mathcal{A}$. The agent then selects an action $a_t \in \mathcal{A}_t$ and receives randomized reward $r_t = f_*(x_t, a_t) + \epsilon_t$, where $f_* : \mathcal{X} \times \mathcal{A}_t \to [0, 1]$ is the ground truth reward function which is unknown to the agent, and $\epsilon_t$ is the zero-mean noise. We use $\mathcal{F}_t$ to denote the filtration generated by $\{(x_t, a_t, f_t, r_t)\}_{t \in [T]}$, i.e., all the randomness up to step $t$ (including the randomness if the reward function $f_t$ is sampled), and $\mathcal{G}_t$ to denote the filtration that includes all the randomness of $\mathcal{F}_t$ but $r_t$. The goal of the agent is to minimize the total regret defined as

$$\text{Regret}(T) = \sum_{t=1}^{T} [f_*(x_t, a_t^*) - f_*(x_t, a_t^*)],$$

where $a_t^* = \text{argmax}_{a \in \mathcal{A}_t} f_*(x_t, a)$ is the optimal action in step $t$.

**Subgaussian noise.** Following previous works on FGTS for contextual bandits (Zhang, 2022; Fan and Gu, 2023), we assume that the noise is subgaussian, and denote the subgaussian norm of $\epsilon_t$ as $\sigma_t^2$ which is formalized by the following assumption:

**Assumption 3.1.** The noise $\epsilon_t$ is $\sigma_t^2$-subgaussian conditioned on the history, i.e., for any $\lambda$, the moment-generating function of $\epsilon_t$ conditioned on $\mathcal{G}_t$ satisfies

$$\log \mathbb{E}[\exp(\lambda \epsilon_t) | \mathcal{G}_t] \leq \sigma_t^2 \lambda^2 / 8.$$

We denote $\Lambda := \sum_{t=1}^{T} \sigma_t^2$. We assume that $\sigma_t^2$ is revealed to the agent at the beginning of step $t$, which is referred to as the setting of "weak adversary with variance revealing" in Jia et al. (2024). Compared with the assumption used in UCB-based variance-aware algorithms (Zhou et al., 2021; Kim et al., 2022; Zhou and Gu, 2022; Zhao et al., 2023; Xu et al., 2023) where $\epsilon_t$ is bounded with variance $\mathbb{E}[\epsilon_t^2 | \mathcal{G}_t]$, Assumption 3.1 subsumes that $\mathbb{E}[\epsilon_t^2 | \mathcal{G}_t] = \mathcal{O}(\sigma_t^2)$ (but not the boundedness of $\epsilon_t$) because

$$\mathbb{E}[\epsilon_t^2 | \mathcal{G}_t] = \lim_{\lambda \to 0} \frac{\mathbb{E}[\exp(\lambda \epsilon_t) | \mathcal{G}_t] - 1}{\lambda^2 / 2} \leq \lim_{\lambda \to 0} \frac{\exp(\sigma_t^2 \lambda^2 / 8) - 1}{\lambda^2 / 2} = \frac{\sigma_t^2}{4}.$$

**Reward function class.** The agent may estimate the reward function that belongs to a function class $\mathcal{F}$. We focus on the realizable setting where the ground truth reward function satisfies $f_* \in \mathcal{F}$. An example of the reward function class is the family of linear functions denoted by $\mathcal{F}_d^{\text{lin}} = \{f_{\boldsymbol{\theta}} : \boldsymbol{\theta} \in \Theta\}$, where $f_{\boldsymbol{\theta}}(x, a) = \langle \boldsymbol{\theta}, \phi(x, a) \rangle \in [0, 1]$, $\Theta \subset \{\mathbf{w} \in \mathbb{R}^d : \|\mathbf{w}\|_2 \leq 1\}$ is the parameter space, and $\phi : \mathcal{X} \times \mathcal{A} \to \mathbb{R}^d$ is a feature mapping. We also consider the function class with finite generalized Eluder dimension (Agarwal et al., 2023), defined as follows:

**Definition 3.2.** Let $Z = \{z_t\}_{t \in [T]}$ be a sequence of context-action pairs, and $\boldsymbol{\beta} = \{\beta_t\}_{t \in [T]}$ be positive numbers. The generalized Eluder dimension of the function class $\mathcal{F}$ is given by $\dim_{\lambda, \epsilon, T}(\mathcal{F}) := \sup_{Z, \boldsymbol{\beta} \leq \epsilon} \dim_\lambda(\mathcal{F}, Z, \boldsymbol{\beta})$, where

$$\dim_\lambda(\mathcal{F}, Z, \boldsymbol{\beta}) = \sum_{t=1}^{T} \min\{1, \beta_t \mathcal{D}_{\lambda, \mathcal{F}}^2(z_t, z_{[t-1]}, \boldsymbol{\beta}_{[t-1]})\},$$

$$\mathcal{D}_{\lambda, \mathcal{F}}^2(z, z_{[t-1]}, \boldsymbol{\beta}_{[t-1]}) = \sup_{f_1, f_2 \in \mathcal{F}} \frac{(f_1(z) - f_2(z))^2}{\lambda + \sum_{s=1}^{t-1} \beta_s (f_1(z_s) - f_2(z_s))^2}.$$

### 3.2 Feel-Good Thompson Sampling

In Thompson Sampling (Thompson, 1933), instead of deterministically estimating the reward function, an estimation of the reward function $f_t$ is sampled in step $t$ from the posterior distribution defined as

$$p_t^{\text{TS}}(f | S_{t-1}) \propto p_0(f) \exp\left(-\sum_{s=1}^{t-1} L_s(f, x_s, a_s, r_s)\right), \tag{3.1}$$

where $p_0(f)$ is a prior distribution, and $L_s$ is the log-likelihood often set as the square loss $L_s(f, x_s, a_s, r_s) = \eta(r_s - f(x_s, a_s))^2$ with $\eta$ being a hyperparameter. An action is then selected from $\mathcal{A}_t$ to maximize $f_t(x_t, \cdot)$. However, Zhang (2022) showed that the frequentist regret of vanilla Thompson sampling is suboptimal in the worst case, and proposed Feel-Good Thompson sampling (FGTS) to fill this gap. At the core of FGTS is a modification to the posterior distribution called the **feel-good exploration** term of the form $\max_{a \in \mathcal{A}_t} f(x_t, a)$. Two types of feel-good exploration terms have been developed:

Type A. Augment the log-likelihood of each step with the feel-good exploration term (Zhang, 2022; Agarwal and Zhang, 2022), i.e.,

$$p_t^{\text{FGTS}-\text{A}}(f|S_{t-1}) \propto p_0(f) \exp \left( \sum_{s=1}^{t-1} [-\eta(r_s - f(x_s, a_s))^2 + \lambda \max_{a \in \mathcal{A}_s} f(x_s, a)] \right). \tag{3.2}$$

Type B. Only add the feel-good exploration term of the current step (Dann et al., 2021), i.e.,

$$p_t^{\text{FGTS}-\text{B}}(f|S_{t-1}) \propto p_0(f) \exp \left( -\sum_{s=1}^{t-1} \eta(r_s - f(x_s, a_s))^2 + \lambda \max_{a \in \mathcal{A}_t} f(x_t, a) \right). \tag{3.3}$$

Different from (Zhang, 2022) that applied Type A of FGTS to contextual bandits, our algorithm is more closely related to Type B. We will present details of our posterior distribution in Section 4, show the effectiveness of Type B applied to (variance-aware) contextual bandits in Section 5 (although it is originally developed for model-free RL), and explain the reason why Type B is preferred for variance-aware FGTS in Section 6.

## 4   Variance Aware Feel-Good Thompson Sampling

In this section, we sketch our algorithm `FGTS-VA` in Algorithm 1 for contextual bandits with heterogeneous noise levels. `FGTS-VA` adopts the general framework of FGTS algorithms where an estimation of the reward function $f_t$ is sampled from the posterior distribution augmented with the feel-good exploration term, and then the action $a_t$ is selected to maximize the reward function.

---
**Algorithm 1** `FGTS-VA`
---
1: Given hyperparameter $\alpha$ and $\gamma$. Initialize $S_0 = \varnothing$.
2: **for** $t = 1$ **to** $T$ **do**
3:     Receive context $x_t$.
4:     Set parameters $\{\eta_s\}_{s \in [t-1]}$ and $\lambda_t$ according to (4.2).
5:     Sample $f_t \sim p_t(\cdot|S_{t-1})$, with the posterior distribution $p_t(f|S_{t-1})$ defined in (4.1).
6:     Select $a_t = \text{argmax}_{a \in \mathcal{A}_t} f_t(x_t, a)$.
7:     Observe reward $r_t$; update $S_t = S_{t-1} \cup \{(x_t, a_t, r_t)\}$.
8: **end for**
---

**Posterior distribution.** Motivated by (3.3), the posterior distribution is designed as

$$p_t(f|S_{t-1}) \propto p_0(f) \exp \left( -\sum_{s=1}^{t-1} \eta_s (r_s - f(x_s, a_s))^2 + \lambda_t \max_{a \in \mathcal{A}_t} f(x_t, a) \right), \tag{4.1}$$

where parameters $\eta_s$ and $\lambda_t$ are chosen as

$$\eta_s = \bar{\sigma}_s^{-2}, \quad \lambda_t = c\sqrt{\Lambda_t}/\bar{\sigma}_t^2, \quad \text{where } \bar{\sigma}_t = \max\{\sigma_t, \alpha\}, \quad \Lambda_t = \sum_{s=1}^{t} \bar{\sigma}_s^2. \tag{4.2}$$

with $\alpha$ and $c$ being hyperparameters. We explain the design of $\eta_t$ and $\lambda_t$ as follows:

- The constructions of $\eta_s$ and $\bar{\sigma}_s$ are similar to Zhou et al. (2021), with $\alpha > 0$ being a hyperparameter that controls $\eta_s$ in case of vanishing $\sigma_s^2$ and can be set as $\mathcal{O}(1/\text{poly}(T))$. The coefficient $\eta_s$ functions as a preconditioner that balances the squared error $(r_s - f(x_s, a_s))^2$ across different steps. Compared with our algorithm, Zhou and Gu (2022) and Xu et al. (2023) set $\bar{\sigma}_t$ as the maximum of not only $\sigma_t$ and $\alpha$, but also a quantity called the "uncertainty" that depends on the action $a_t$. However, this approach is prohibitive in our algorithm due to the occurrence of $\bar{\sigma}_t$ in $\lambda_t$ which is required before the choice of $a_t$.

- The parameter $\lambda_t$ controls the magnitude of the feel-good exploration term. $\lambda_t$ scales with $\bar{\sigma}_t^{-2}$ because intuitively, if the reward of the current step is small, then exploration should be encouraged due to the more informative reward feedback. Setting $\lambda_t = c\sqrt{\Lambda_t}/\bar{\sigma}_t^2$ achieves the same regret as setting $\lambda_t = c\sqrt{\Lambda}/\bar{\sigma}_t^2$ (explained in Section 6), and avoids requiring the total variance $\Lambda$ at initialization.

## 5 Main Results

We first introduce the generalized decoupling coefficient, an extension of the standard decoupling coefficient in Dann et al. (2021) which is a crucial tool in the analysis of algorithms based on FGTS:

**Definition 5.1** (Generalized decoupling coefficient). Let $Z = \{z_t\}_{t \in [T]}$ be a sequence of context-action pairs, and $\boldsymbol{\beta} = \{\beta_t\}_{t \in [T]}$ be positive numbers. The decoupling coefficient is defined as $\mathrm{dc}_{\lambda,\epsilon,T}(\mathcal{F}) := \sup_{Z, \boldsymbol{\beta} \leq \epsilon} \mathrm{dc}_\lambda(\mathcal{F}, Z, \boldsymbol{\beta})$, where $\mathrm{dc}_\lambda(\mathcal{F}, Z, \boldsymbol{\beta})$ is the smallest number that satisfies

$$\sum_{t=1}^{T}(f_t(z_t) - f_*(z_t)) \leq \sum_{t=1}^{T} \frac{\gamma}{\beta_t} \sum_{s=1}^{t-1} \beta_s (f_t(z_s) - f_*(z_s))^2 + \gamma\lambda \sum_{t=1}^{T} \frac{1}{\beta_t} + \left(1 + \frac{1}{4\gamma}\right) \mathrm{dc}_\lambda(\mathcal{F}, Z, \boldsymbol{\beta}),$$
(5.1)

for any sequence $\{f_t\}_{t \in [T]}$ where $f_t \in \mathcal{F}$ and any $\gamma > 0$.

The generalized decoupling coefficient is used to analyze $f_t(x_t, a_t) - f_*(x_t, a_t)$ where $a_t$ and $f_t$ are dependent random variables because $a_t$ is selected to maximize $f_t(x_t, \cdot)$ in the algorithm design. It relates the error of the current step to the error of historic steps $f_t(z_s) - f_*(z_s)$, and generalizes the standard decoupling coefficient (Dann et al., 2021) by introducing undermined parameters $\beta_t$. We discuss the relationships of the generalized decoupling coefficient with the standard decoupling coefficient and with other complexity measures as follows:

**Relationship with standard decoupling coefficient.** By fixing $\beta_t = 1$, the generalized decoupling coefficient is closely related to the standard decoupling coefficient in Dann et al. (2021): For any $\gamma \leq 1$, the standard decoupling coefficient is defined as $\mathrm{dc}'_T(\mathcal{F}) = \sup_Z \mathrm{dc}'(\mathcal{F}, Z)$, where $\mathrm{dc}'(\mathcal{F}, Z)$ is the smallest number that satisfies

$$\sum_{t=1}^{T}(f_t(z_t) - f_*(z_t)) \leq \gamma \sum_{t=1}^{T} \sum_{s=1}^{T}(f_t(z_s) - f_*(z_s))^2 + \frac{\mathrm{dc}'(\mathcal{F}, Z)}{4\gamma}.$$
(5.2)

Comparing (5.1) and (5.2), two additional terms occur in the definition of the generalized decoupling coefficient: (i) The term $\gamma\lambda \sum_{t=1}^{T} \beta_t^{-1}$ is a technical artifact and can be shaved by choosing small $\lambda$; (ii) The term $\mathrm{dc}_\lambda(\mathcal{F}, Z, \boldsymbol{\beta})$ is shadowed by $\mathrm{dc}_\lambda(\mathcal{F}, Z, \boldsymbol{\beta})/(4\gamma)$ when $\gamma \leq 1$ as is the case in (5.2). However, as more flexible choices of both $\boldsymbol{\beta}$ and $\gamma$ (possibly $> 1$) are required, this additional term is unavoidable in the variance-aware setting. The occurrence of this term is due to the possibly large values of $\beta_t$, and will be explained in detail in Appendix A (see (A.1)).

**Relationship with other complexity measures.** For the linear reward function class, we can upper bound the generalized reward function as follows:

**Proposition 5.2.** For the linear function class $\mathcal{F}_d^{\mathrm{lin}}$, the generalized decoupling coefficient satisfies

$$\mathrm{dc}_{\lambda,\epsilon,T}(\mathcal{F}_d^{\mathrm{lin}}) \leq 2d\log(1 + (\epsilon T)/(d\lambda)).$$

For the general reward function class, the generalized decoupling coefficient can be bounded by the generalized Eluder dimension (Agarwal et al., 2023) as follows:

**Proposition 5.3.** For a reward function class with finite generalized Eluder dimension, the generalized decoupling coefficient satisfies $\mathrm{dc}_{\lambda,\epsilon,T}(\mathcal{F}) \leq \dim_{\lambda,\epsilon,T}(\mathcal{F})$.

The proofs of Propositions 5.2 and 5.3 are given in Appendix A.

We now present the regret upper bound of FGTS-VA:

**Theorem 5.4.** Suppose that Assumption 3.1 holds, the function class $\mathcal{F}$ has finite cardinality, and the prior distribution $p_0$ is the uniform distribution on $\mathcal{F}$. Assume that parameters $\eta_t$ and $\lambda_t$ are chosen according to (4.2), and the hyperparameters are chosen as

$$\alpha = 1/\sqrt{T}, \quad \lambda = 1, \quad \epsilon = \alpha^{-2}, \quad c = 2\sqrt{\mathrm{dc}_{\lambda,\epsilon,T}^{-1}(\mathcal{F})\log|\mathcal{F}|}.$$

Then the total regret of `FGTS-VA` satisfies

$$\mathbb{E}[\text{Regret}(T)] \lesssim \sqrt{(1 + \Lambda)\,\text{dc}_{\lambda,\epsilon,T}(\mathcal{F})\log|\mathcal{F}|} + \text{dc}_{\lambda,\epsilon,T}(\mathcal{F}).$$

The proof of Theorem 5.4, as well as a more general version of Theorem 5.4 that extends to the case of infinite function class, is given in Appendix B.

**Remark 5.5.** Under the setting of linear contextual bandits, $\mathcal{F}$ can be chosen as the $\varepsilon$-net of the unit ball whose cardinality satisfies $\log|\mathcal{F}| = \widetilde{\mathcal{O}}(d)$. Additionally, the generalized decoupling coefficient satisfies $\text{dc} = \widetilde{\mathcal{O}}(d)$ as is shown in Proposition 5.2. Therefore, when applied to linear contextual bandits, `FGTS-VA` has a **nearly-optimal regret** of $\widetilde{\mathcal{O}}(d\sqrt{\Lambda} + d)$, similar to UCB-based algorithms (Zhou and Gu, 2022; Zhao et al., 2023).

**Remark 5.6.** Under the setting of deterministic reward, the total variance is $\Lambda = 0$, and the regret of `FGTS-VA` is $\mathcal{O}(\text{dc})$[1]. Note that the generalized decoupling coefficient is upper bounded by the generalized Eluder dimension, which reduces to the standard Eluder dimension (Russo and Van Roy, 2013) in the deterministic case. Therefore, the regret of `FGTS-VA` is **minimax-optimal** in the deterministic case for the general reward function class (Jia et al., 2024).

**Remark 5.7.** When $\sigma_t^2 = 1$ for all $t \in [T]$, the setting reduces to the standard contextual bandits, and the reward of `FGTS-VA` is $\widetilde{\mathcal{O}}(\sqrt{T\,\text{dc}\cdot\log|\mathcal{F}|} + \text{dc})$, which is $\widetilde{\mathcal{O}}(d\sqrt{T})$ for linear contextual bandits. Therefore, although Dann et al. (2021) only studied FGTS in model-free RL, we have shown that a similar posterior distribution in (3.3), is applicable to the setting of contextual bandits, and enjoys the minimax-optimal regret bound for the linear regret function class similar to Zhang (2022). Additionally, since the parameters are $\eta_t = 1$ and $\lambda_t = \Theta(\text{dc}^{-1}\sqrt{t}\log|\mathcal{F}|)$, `FGTS-VA` is reduced to the **first FGTS-based algorithm that does not require knowledge of the horizon** $T$.

## 6  Overview of Proof

We first define several shorthand notations: We use $\text{dc}$ to denote the (generalized) decoupling coefficient, and denote

$$\Delta L(f, x, a, r) = (r - f(x, a))^2 - (r - f_*(x, a))^2, \quad \text{FG}_t(f) = \max_{a \in \mathcal{A}_t} f(x_t, a) - f_*(x_t, a_t^*),$$

$$\text{LS}_t(f) = (f(x_t, a_t) - f_*(x_t, a_t))^2,$$

then the posterior distribution (4.1) is equivalent to

$$p(f|S_{t-1}) \propto p_0(f)\exp\left(-\sum_{s=1}^{t-1}\Delta L(f, x_s, a_s, r_s) + \lambda_t\,\text{FG}_t(f)\right). \tag{6.1}$$

Note that the regret at step $t$ can be decomposed as

$$\mathbb{E}[f_*(x_t, a_t^*) - f_*(x_t, a_t)] = \mathbb{E}[f_t(x_t, a_t) - f_*(x_t, a_t)] - [f_t(x_t, a_t) - f_*(x_t, a_t^*)]$$
$$= \underbrace{\mathbb{E}[f_t(x_t, a_t) - f_*(x_t, a_t)]}_{\text{Bellman Error}} - \mathbb{E}[\text{FG}_t(f_t)],$$

where the second equality holds because $a_t$ is the maximizer of $f_t(x_t, \cdot)$. The Bellman error term $\mathbb{E}[f_t(x_t, a_t) - f_*(x_t, a_t)]$ is then bounded using the (generalized) decoupling coefficient, which has two versions corresponding to the two types of posterior distributions. In the remaining of this section, we will explain the reason why Type B of the posterior distribution in (3.3) is the basis of our algorithm instead of Type A in (3.2) by first explaining the obstacles in the analysis based on Type A, and then showing how `FGTS-VA` built on Type B manages to overcome the obstacle. We will also explain the technical trick in the construction of $\lambda_t$.

### 6.1  Technical Obstacle when Applying Type A of FGTS

When applying the posterior distribution similar to (3.2), i.e.

$$p_t(f|S_{t-1}) \propto p_0(f)\exp\left(\sum_{s=1}^{t-1}[-\eta_s\Delta L(f, x_s, a_s, r_s) + \lambda_s\,\text{FG}_s(f)]\right),$$

---

[1]Although there is a $1 + \Lambda$ term in the regret bound of Theorem 5.4, it can be further suppressed by setting $\alpha$ to be an even smaller number.

the proof developed based on Zhang (2022) requires that $\eta_t$ is upper bounded by an absolute constant that is irrelevant to $T$, which results in the regret bound polynomial in $T$ because of the occurrence of $\sum 1/\eta_t$ in the regret bound. In detail, by applying the decoupling coefficient for this type of posterior distribution, we have

$$\mathbb{E}[f(x_t, a_t) - f_*(x_t, a_t)] \leq \frac{\mathrm{dc}}{4\gamma_t} + \gamma_t \mathbb{E}_{S_{t-1}, x_t} \mathbb{E}_{a_t | S_{t-1}, x_t} \mathbb{E}_{\widetilde{f} \sim p_t} \mathrm{LS}_t(\widetilde{f}).$$

Therefore, it suffices to prove an upper bound for

$$\sum_{t=1}^{T} \frac{\mathrm{dc}}{4\gamma_t} + \sum_{t=1}^{T} \mathbb{E}_{S_{t-1}, x_t} \mathbb{E}_{\widetilde{f} \sim p_t} \left[ \gamma_t \mathbb{E}_{a_t | S_{t-1}, x_t} \mathrm{LS}_t(\widetilde{f}) - \mathrm{FG}_t(\widetilde{f}) \right]. \tag{6.2}$$

We define the potential as

$$Z_t = \mathbb{E}_{S_t} \log \mathbb{E}_{f \sim p_0} \exp \left( \sum_{s=1}^{t} [-\eta_s \Delta L(f, x_s, a_s, r_s) + \lambda_s \mathrm{FG}_s(f, x_s)] \right).$$

The proof proceeds by bounding $Z_t - Z_{t-1}$ and applying the telescope sum. Note that

$$Z_t - Z_{t-1} = \mathbb{E}_{S_t} \log \mathbb{E}_{\widetilde{f} \sim p_t} \exp \left( -\eta_s \Delta L(f, x_t, a_t, r_t) + \lambda_t \mathrm{FG}_t(f, x_s) \right)$$

$$\leq \frac{1}{2} \mathbb{E}_{S_t} \Big[ \underbrace{\log \mathbb{E}_{\widetilde{f} \sim p(\cdot|S_{t-1})} \exp(-2\eta_s \Delta L(f, x_t, a_t, r_t))}_{I_1} + \underbrace{\log \mathbb{E}_{\widetilde{f} \sim p(\cdot|S_{t-1})} \exp(2\lambda_t \mathrm{FG}_t(\widetilde{f}, x_t))}_{I_2} \Big],$$

where the inequality holds due to Hölder's inequality. By using the Hoeffding's Lemma, the term $I_2$ can be bounded by $2\lambda_t \mathbb{E}_{\widetilde{f} \sim p(\cdot|S_{t-1})} \mathrm{FG}_t(\widetilde{f}, x_t) + 2\lambda_t^2$. For the term $I_1$, By first taking the conditional expectation on $\mathcal{G}_t$ and using Assumption 3.1, we have

$$\mathbb{E}[I_1 | \mathcal{G}_t] \leq \log \mathbb{E}_{\widetilde{f} \sim p_t} \exp(-2\eta_t(1 - \sigma_t^2 \eta_t) \mathrm{LS}_t(\widetilde{f})). \tag{6.3}$$

By choosing $\eta_t \leq \sigma_t^{-2}/2$, the coefficient $2\eta_t(1 - \sigma_t^2 \eta_t)$ can be lower bounded by $\eta_t$. To connect (6.3) with (6.2), the RHS of (6.3) has to be bounded by $-C\eta_t \cdot \mathbb{E}_{\widetilde{f} \sim p(\cdot|S_{t-1})} \mathrm{LS}_t(\widetilde{f})$ where $C$ is an absolute constant. This is possible only when $\eta_t \mathrm{LS}_t(\widetilde{f}) = \mathcal{O}(1)$. Since $\mathrm{LS}_t(\widetilde{f}) = \Theta(1)$ in the worst case, $\eta_t$ should be an absolute constant. Comparing what we obtain against (6.2), we note that (i) $\lambda_t$ has to be a constant $\lambda$ to enable the telescope sum of $Z_t - Z_{t-1}$, and (ii) to make coefficients match, we require $\gamma_t = C\eta_t/\lambda = \Theta(\lambda^{-1})$. Thus, the first term of (6.2) becomes $\mathcal{O}(\lambda T \mathrm{dc})$, with an undesirable factor of $T$. Therefore, Type A of FGTS in (3.2) cannot yield variance-aware regret bounds with existing techniques, even if inhomogeneous parameters $\eta_t$ and $\lambda_t$ are allowed.

## 6.2 Highlight of Proof Techniques

By proceeding through a sharply different path, the analysis of `FGTS-VA` avoids the aforementioned obstacle that stems from bounding the expectation of the exponential term on RHS of (6.3). The following two techniques work together to relate the posterior distribution with the desired form of the decoupling coefficient:

**Technique 1: Prioritizing expectation over the randomness of reward.** Although the expectation of an exponential term over the randomness of posterior sampling causes trouble, the expectation over the randomness of $\epsilon_t$ adopts a simple form due to Assumption 3.1. Therefore, by defining

$$\xi_s(\widetilde{f}, x_s, a_s, r_s) = -\eta_s \Delta L(\widetilde{f}, x_s, a_s, r_s) - \log \mathbb{E}[\exp(-\eta_s \Delta L(\widetilde{f}, x_s, a_s, r_s)) | \mathcal{G}_s],$$

we have the following property (following Dann et al. (2021); formalized in Lemma D.1):

$$\mathbb{E}_{S_t} \exp \left( \sum_{s=1}^{t} \xi_s(\widetilde{f}, x_s, a_s, r_s) \right) = 1.$$

By using the Jensen's inequality, we have

$$0 = \log \mathbb{E}_{\widetilde{f} \sim p_0} \mathbb{E}_{S_{t-1}, x_t} \exp \left( \sum_{s=1}^{t-1} \xi_s(\widetilde{f}, x_s, a_s, r_s) \right)$$

$$\geq \mathbb{E}_{S_{t-1}, x_t} \log \mathbb{E}_{\widetilde{f} \sim p_0} \exp \left( \sum_{s=1}^{t-1} \xi_s(\widetilde{f}, x_s, a_s, r_s) \right). \tag{6.4}$$

**Technique 2: KL-regularized optimality.** The following lemma is used to remove the exponential on the RHS of (6.4):

**Lemma 6.1** (Donsker–Varadhan duality, see e.g., Proposition 7.16 in Zhang (2023)). *Let $(\mathcal{X}, \mathcal{F}, P_0)$ be a probability space and $U(x)$ be a measurable function. Then for any distribution $P$ on $(\mathcal{X}, \mathcal{F})$, we have*

$$\mathbb{E}_{x \sim P}[U(x)] + \mathrm{KL}(P \| P_0) \geq -\log \mathbb{E}_{x \sim P_0} \exp(-U(x)),$$

*and the infimum is attained when $P(x) \propto P_0(x) \exp(-U(x))$.*

The RHS of Lemma 6.1 contains the expectation of the exponential term, similar to the RHS of (6.4), and the LHS is the simple expectation of $U(\cdot)$, free of exponential terms. The price of the removal of the exponential is an additional KL-divergence term, so we use Lemma 6.1 twice to cancel out the KL-divergence, one using the inequality itself, and the other using the optimality condition:

$$\log \mathbb{E}_{\widetilde{f} \sim p_0} \exp \left( \sum_{s=1}^{t-1} \xi_s(\widetilde{f}, x_s, a_s, r_s) \right) \geq \sum_{s=1}^{t-1} \mathbb{E}_{\widetilde{f} \sim p_t} \xi_s(\widetilde{f}, x_s, a_s, r_s) - \mathrm{KL}(p_t \| p_0); \tag{6.5}$$

$$-\mathrm{KL}(\delta_{f_*} \| p_0) \leq \mathbb{E}_{\widetilde{f} \sim p_t} \left[ -\sum_{s=1}^{t-1} \eta_s \Delta L(\widetilde{f}, x_s, a_s, r_s) + \lambda_t \, \mathrm{FG}_t(\widetilde{f}) \right] - \mathrm{KL}(p_t \| p_0). \tag{6.6}$$

Plugging (6.5) and (6.6) into (6.4), noting that $\mathrm{KL}(\delta_{f_*} \| p_0) = \log |\mathcal{F}|$, we have

$$\log |\mathcal{F}| \geq \mathbb{E}_{S_{t-1}, x_t} \mathbb{E}_{\widetilde{f} \sim p_t} \left[ -\sum_{s=1}^{t-1} \log \mathbb{E}[\exp(-\eta_s \Delta L(\widetilde{f}, x_s, a_s, r_s) | \mathcal{G}_s] - \lambda_t \, \mathrm{FG}_t(\widetilde{f}) \right]$$

$$\geq \mathbb{E}_{S_{t-1}, x_t} \mathbb{E}_{\widetilde{f} \sim p_t} \left[ \sum_{s=1}^{t-1} \frac{\eta_s}{2} \mathrm{LS}_s(\widetilde{f}) - \lambda_t \, \mathrm{FG}_t(\widetilde{f}) \right], \tag{6.7}$$

where the second inequality holds due to Assumption 3.1 with an argument similar to (6.3). (6.7) is thus completely free of the expectation of exponential terms.

**Avoiding $\Lambda$ in parameters.** If we follow the proof of Dann et al. (2021), then we require $\lambda_t = 1/(2\gamma \bar{\sigma}_t^2)$ where $\gamma$ scales with $\Lambda^{-0.5}$, which is unknown to the agent. To resolve this issue, we observe that the proof can proceed by replacing the total variance $\Lambda$ with the partial sum $\Lambda_t$. Specifically, dividing $\lambda_t = c\sqrt{\Lambda_t}/\bar{\sigma}_t^2$ on both sides of the (6.7), noting that $\Lambda_t \leq \Lambda_T$, we have

$$\frac{\bar{\sigma}_t^2}{c\sqrt{\Lambda_t}} \log |\mathcal{F}| \geq \mathbb{E}_{S_{t-1}, x_t} \mathbb{E}_{\widetilde{f} \sim p_t} \left[ \frac{\bar{\sigma}_t^2}{2c\sqrt{\Lambda_T}} \sum_{s=1}^{t-1} \bar{\sigma}_s^{-2} \mathrm{LS}_s(\widetilde{f}) - \mathrm{FG}_t(\widetilde{f}) \right].$$

Plugging the inequality into the definition of the decoupling coefficient, we have

$$\mathbb{E}[\mathrm{Regret}(T)] \lesssim \frac{\log |\mathcal{F}|}{c} \sum_{t=1}^{T} \frac{\bar{\sigma}_t^2}{\sqrt{\Lambda_t}} + \left( 1 + \frac{c\sqrt{\Lambda_T}}{2} \right) \mathrm{dc} \leq \frac{2\sqrt{\Lambda_T} \log |\mathcal{F}|}{c} + \left( 1 + \frac{c\sqrt{\Lambda_T}}{2} \right) \mathrm{dc},$$

where we use the crucial technical lemma (Lemma D.2) stating that $\sum_{t=1}^{T} \bar{\sigma}_t^2/\sqrt{\Lambda_t} \leq 2\sqrt{\Lambda_T}$. Finally, by choosing $c = 2\sqrt{\mathrm{dc}^{-1} \log |\mathcal{F}|}$ (which is irrelevant to $\Lambda$) and noting that $\Lambda_T = \Theta(\alpha^2 T + \Lambda)$, the regret can be bounded by $\mathcal{O}(\sqrt{(\alpha^2 T + \Lambda) \, \mathrm{dc} \cdot \log |\mathcal{F}|} + \mathrm{dc})$.

# 7 Experiments

In this section, we examine our algorithm, `FGTS-VA`, against baselines (including Weighted OFUL+, FGTS, and SAVE) in experiments with synthetic data. The code can be found at `https://github.com/xuheng-li99/FGTS-VA`.

**Environment.** We focus on the setting of linear bandits with $d = 5$ and $\mathcal{X} = \{x\}$, so we omit the context $x$ for simplicity. The action set is $\mathcal{A}_t = \mathcal{A} = \{\pm 1/\sqrt{d}\}^d$, and the ground truth parameter $\boldsymbol{\theta}_*$ is sampled from the uniform distribution on the unit sphere. We consider two noise models with heterogeneous noise magnitudes. In both cases, the noise $\epsilon_t$ is sampled from $\mathcal{N}(0, \sigma_t^2)$.

1. The noise is sparse: $\sigma_t^2 = 1$ with probability $p$, and $\sigma_t^2 = 0$ with probability $1 - p$. We set $p = 0.1$ in our experiments.

2. The noise is dense: $\sigma_t^2$ is sampled from a $\chi^2$ distribution with degree of freedom equal to 1.

**Implementation details.** For `FGTS-VA`, in the linear bandit setting, we let the prior distribution be the Gaussian distribution $\mathcal{N}(\mathbf{0}, \mathbf{I}_d/d)$. We use Langevin dynamics to sample from this distribution:

$$\boldsymbol{\theta}_t^{(k+1)} = \boldsymbol{\theta}_t^{(k)} + \delta^{(k)} \nabla \log p(\boldsymbol{\theta}|S_{t-1}) + \sqrt{2\delta^{(k)}} \boldsymbol{\epsilon}_t,$$

where $\boldsymbol{\epsilon}_t$ is the standard Gaussian noise, and $\delta^{(k)}$ is the stepsize. We use $K = 20$ SGLD steps in our experiments, and initialize $\boldsymbol{\theta}_{t+1}^{(0)} = \boldsymbol{\theta}_t^{(K)}$.

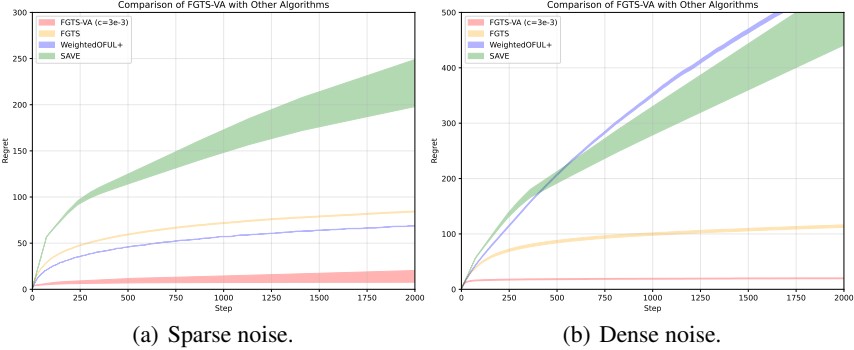

(a) Sparse noise.  (b) Dense noise.

Figure 1: Comparison of different algorithms. Error bands are plotted over 100 runs.

**Comparison of different algorithms.** We first compare `FGTS-VA` with $c = 0.003$ against Weighted OFUL+ (Zhou and Gu, 2022), SAVE (Zhao et al., 2023), and `FGTS` (Zhang, 2022) with results in Figure 1. For both data models, `FGTS-VA` outperforms the baselines by a large margin.

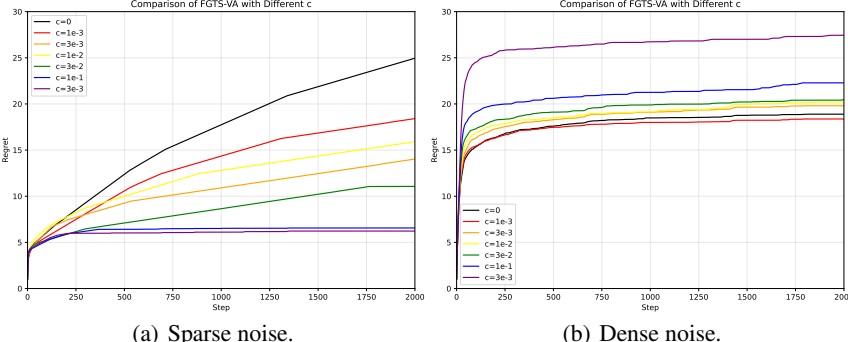

(a) Sparse noise.  (b) Dense noise.

Figure 2: Comparison of different choices of $c$. The averages of regret over 100 runs are plotted.

**Ablation studies.** We then perform ablation studies of the algorithm with different choices of $c$. It is worth noting that $c$ is the only tunable parameter of `FGTS-VA`, and $c = \widetilde{\Theta}(1)$ for linear bandits according to Theorem 5.4. The results are shown in Figure 2. For the case of sparse noise, we observe the advantage of choosing $c$ bounded away from 0, i.e., advantage of the feel-good exploration. For the case of dense noise, the optimal choice of $c$ is close to 0.

# 8 Conclusion

In this work, we present `FGTS-VA`, a variance-aware algorithm for general contextual bandits based on Feel-Good Thompson sampling. In the posterior distribution, we incorporate not only variance-related weights, but also a feel-good exploration term that adopts the idea from model-free RL (Dann et al., 2021). The generalized decoupling coefficient is the pivotal technique in our analysis, with which we show that `FGTS-VA` achieves a nearly optimal regret bound similar to UCB-based algorithms. A restriction of our work is the setting of variance revealing, so it is interesting to explore the possibility of designing FGTS-based algorithms without requiring the variance of the current step as in Zhou and Gu (2022), and ultimately without knowing the variance at all similar to Zhao et al. (2023). Extending of the techniques in this work to reinforcement learning is also an interesting future direction.

## Acknowledgment

We thank the anonymous reviewers and area chair for their helpful comments. XL and QG are supported in part by the National Science Foundation DMS-2323113 and IIS-2403400. The views and conclusions contained in this paper are those of the authors and should not be interpreted as representing any funding agencies.

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

# A  Generalized Decoupling Coefficient

## A.1  Linear Reward Function Class

In this section, we prove the decoupling coefficient for linear contextual bandits. We assume that $\boldsymbol{\theta}_* = \mathbf{0}$ without loss of generality. We use the shorthand notation $\boldsymbol{\phi}_t := \boldsymbol{\phi}(x_t, a_t)$, and define

$$\boldsymbol{\Sigma}_t := \lambda \mathbf{I} + \sum_{s=1}^{t} \beta_s \boldsymbol{\phi}_s \boldsymbol{\phi}_s^\top.$$

The following lemma, known as the elliptical potential lemma, is a well-known result in previous works (Abbasi-Yadkori et al., 2011; Zhou and Gu, 2022):

**Lemma A.1.** For any $\lambda > 0$, we have

$$\sum_{t=1}^{T} \min\left\{\beta_t \|\boldsymbol{\phi}_t\|_{\boldsymbol{\Sigma}_{t-1}^{-1}}^2, 1\right\} \le 2(\log \det(\boldsymbol{\Sigma}_T) - \log \det(\boldsymbol{\Sigma}_0)) \le 2d \underbrace{\log(1 + (\epsilon T)/(d\lambda))}_{\iota}.$$

With Lemma A.1, we decompose the sum of $\langle \boldsymbol{\theta}_t, \boldsymbol{\phi}_t \rangle$ into cases based on whether $\beta_t \|\boldsymbol{\phi}_t\|_{\boldsymbol{\Sigma}_{t-1}^{-1}}^2$ is smaller than 1, i.e.,

$$\sum_{t=1}^{T} \langle \boldsymbol{\theta}_t, \boldsymbol{\phi}_t \rangle = \underbrace{\sum_{t=1}^{T} \langle \boldsymbol{\theta}_t, \boldsymbol{\phi}_t \rangle \mathbb{1}\left[\beta_t \|\boldsymbol{\phi}_t\|_{\boldsymbol{\Sigma}_{t-1}^{-1}}^2 \le 1\right]}_{I_1} + \underbrace{\sum_{t=1}^{T} \langle \boldsymbol{\theta}_t, \boldsymbol{\phi}_t \rangle \mathbb{1}\left[\beta_t \|\boldsymbol{\phi}_t\|_{\boldsymbol{\Sigma}_{t-1}^{-1}}^2 > 1\right]}_{I_2}. \tag{A.1}$$

For any $\gamma > 0$, the term $I_1$ satisfies

$$
\begin{aligned}
I_1 &\le \sum_{t=1}^{T} \beta_t^{-1/2} \|\boldsymbol{\theta}_t\|_{\boldsymbol{\Sigma}_{t-1}} \cdot \beta_t^{1/2} \|\boldsymbol{\phi}_t\|_{\boldsymbol{\Sigma}_{t-1}^{-1}} \cdot \mathbb{1}\left[\beta_t \|\boldsymbol{\phi}_t\|_{\boldsymbol{\Sigma}_{t-1}^{-1}}^2 \le 1\right] \\
&\le \sum_{t=1}^{T} \beta_t^{-1/2} \|\boldsymbol{\theta}_t\|_{\boldsymbol{\Sigma}_{t-1}} \cdot \min\left\{\beta_t^{1/2} \|\boldsymbol{\phi}_t\|_{\boldsymbol{\Sigma}_{t-1}^{-1}}, 1\right\} \\
&\le \sum_{t=1}^{T} \frac{\gamma}{\beta_t} \|\boldsymbol{\theta}_t\|_{\boldsymbol{\Sigma}_{t-1}}^2 + \frac{1}{4\gamma} \sum_{t=1}^{T} \min\left\{\beta_t \|\boldsymbol{\phi}_t\|_{\boldsymbol{\Sigma}_{t-1}^{-1}}^2, 1\right\} \\
&\le \sum_{t=1}^{T} \frac{\gamma}{\beta_t} \left(\lambda \|\boldsymbol{\theta}_t\|_2^2 + \sum_{s=1}^{t-1} \beta_s \langle \boldsymbol{\theta}_t, \boldsymbol{\phi}_s \rangle^2\right) + \frac{d\iota}{2\gamma} \\
&\le \gamma \lambda \sum_{t=1}^{T} \frac{1}{\beta_t} + \sum_{t=1}^{T} \frac{\gamma}{\beta_t} \sum_{s=1}^{t-1} \beta_s \langle \boldsymbol{\theta}_t, \boldsymbol{\phi}_s \rangle^2 + \frac{d\iota}{2\gamma}, \tag{A.2}
\end{aligned}
$$

where the first inequality holds due to AM-GM inequality, the second inequality holds because $z \cdot \mathbb{1}[z \le 1] \le \min\{z, 1\}$, the third inequality holds due to Cauchy-Schwarz inequality, the fourth inequality holds due to Lemma A.1, and the last inequality holds because $\|\boldsymbol{\theta}_t\|_2 \le 1$. The term $I_2$ satisfies

$$I_2 \le \sum_{t=1}^{T} \mathbb{1}\left[\beta_t \|\boldsymbol{\phi}_t\|_{\boldsymbol{\Sigma}_{t-1}^{-1}}^2 > 1\right] \le \sum_{t=1}^{T} \min\left\{\beta_t \|\boldsymbol{\phi}_t\|_{\boldsymbol{\Sigma}_{t-1}^{-1}}^2, 1\right\} \le 2d\iota, \tag{A.3}$$

where the first inequality holds because $\min\{|\boldsymbol{\theta}_t, \boldsymbol{\phi}_t|, 1\} \le 1$, the second inequality holds because $\mathbb{1}[z > 1] \le \min\{z, 1\}$, and the last inequality holds due to Lemma A.1. Plugging (A.2) and (A.3) into (A.1), we have

$$\sum_{t=1}^{T} \langle \boldsymbol{\theta}_t, \boldsymbol{\phi}_t \rangle \le \gamma \sum_{t=1}^{T} \sum_{s=1}^{t-1} \frac{\beta_s}{\beta_t} \langle \boldsymbol{\theta}_t, \boldsymbol{\phi}_s \rangle^2 + \gamma \lambda \sum_{t=1}^{T} \frac{1}{\beta_t} + 2d\iota\left(1 + \frac{1}{4\gamma}\right).$$

## A.2 General Reward Function Class

In this section, we relate the generalized decoupling coefficient to the generalized Eluder dimension (Agarwal et al., 2023). The proof is similar to that of the linear reward function. We first make the following decomposition:

$$\sum_{t=1}^{T}(f_t(z_t) - f_*(z_t)) = \underbrace{\sum_{t=1}^{T}(f_t(z_t) - f_*(z_t))\,\mathbb{1}\left[\beta_t^{1/2}\mathcal{D}_{\mathcal{F}}(z_t; z_{[t-1]}, \boldsymbol{\beta}_{[t-1]}) \leq 1\right]}_{I_1}$$

$$+ \underbrace{\sum_{t=1}^{T}(f_t(z_t) - f_*(z_t))\,\mathbb{1}\left[\beta_t\mathcal{D}_{\mathcal{F}}^2(z_t; z_{[t-1]}, \boldsymbol{\beta}_{[t-1]}) > 1\right]}_{I_2}. \qquad (A.4)$$

The term $I_1$ satisfies

$$I_1 = \sum_{t=1}^{T}\frac{\beta_t^{-1/2}(f_t(z) - f_*(z))}{\mathcal{D}_{\mathcal{F}}(z_t; z_{[t-1]}, \boldsymbol{\beta}_{[t-1]})} \cdot \left(\beta_t^{1/2}\mathcal{D}_{\mathcal{F}}(z_t; z_{[t-1]}, \boldsymbol{\beta}_{[t-1]})\,\mathbb{1}[\beta_t^{1/2}\mathcal{D}_{\mathcal{F}}(z_t; z_{[t-1]}, \boldsymbol{\beta}_{[t-1]}) \leq 1]\right)$$

$$\leq \sum_{t=1}^{T}\frac{\beta_t^{-1/2}(f_t(z_t) - f_*(z_t))}{\mathcal{D}_{\mathcal{F}}(z_t; z_{[t-1]}, \boldsymbol{\beta}_{[t-1]})} \cdot \min\left\{\beta_t^{1/2}\mathcal{D}_{\mathcal{F}}(z_t; z_{[t-1]}, \boldsymbol{\beta}_{[t-1]}), 1\right\}$$

$$\leq \sum_{t=1}^{T}\frac{\gamma}{\beta_t} \cdot \frac{(f_t(z_t) - f_*(z_t))^2}{\mathcal{D}_{\mathcal{F}}^2(z_t, z_{[t-1]}; \boldsymbol{\beta}_{[t-1]})} + \frac{1}{4\gamma}\sum_{t=1}^{T}\min\left\{\beta_t\mathcal{D}_{\mathcal{F}}^2(z_t; z_{[t-1]}, \boldsymbol{\beta}_{[t-1]}), 1\right\}$$

$$\leq \sum_{t=1}^{T}\frac{\gamma}{\beta_t}\left(\lambda + \sum_{s=1}^{t-1}\beta_s(f_t(z_s) - f_*(z_s))^2\right) + \frac{\dim(\mathcal{F}, Z, \boldsymbol{\beta})}{4\gamma}$$

$$\leq \sum_{t=1}^{T}\frac{\gamma}{\beta_t}\sum_{s=1}^{t-1}\beta_s(f_t(z_s) - f_*(z_s))^2 + \gamma\lambda\sum_{t=1}^{T}\frac{1}{\beta_t} + \frac{\dim_{\epsilon,T}(\mathcal{F})}{4\gamma} \qquad (A.5)$$

where the first inequality holds because $z\,\mathbb{1}[z \leq 1] \leq \min\{z, 1\}$, the second inequality holds due to AM-GM inequality, the third inequality holds due to the definition of $\mathcal{D}_{\mathcal{F}}^2(z_t, z_{[t-1]}, \boldsymbol{\beta}_{[t-1]})$ and the definition of $\dim(\mathcal{F}, Z, \boldsymbol{\beta})$, and the last inequality holds because $\dim_{\epsilon,T}(\mathcal{F}) = \sup_{Z,\boldsymbol{\beta}\leq\epsilon}\dim(\mathcal{F}, Z, \boldsymbol{\beta})$. The term $I_2$ satisfies

$$I_2 \leq \sum_{t=1}^{T}\min\left\{\beta_t\mathcal{D}_{\mathcal{F}}^2(z_t; z_{[t-1]}, \boldsymbol{\beta}_{[t-1]}), 1\right\} = \dim(\mathcal{F}, Z, \boldsymbol{\beta}) \leq \dim_{\epsilon,T}(\mathcal{F}), \qquad (A.6)$$

where the first inequality holds because $f_t(z_t) - f_*(z_t) \leq 1$, the equality holds due to the definition of $\dim(\mathcal{F}, Z, \boldsymbol{\beta})$, and the last inequality holds because because $\dim_{\epsilon,T}(\mathcal{F}) = \sup_{Z,\boldsymbol{\beta}\leq\epsilon}\dim(\mathcal{F}, Z, \boldsymbol{\beta})$. Plugging (A.5) into (A.6), we have

$$\sum_{t=1}^{T}(f_t(z_t) - f_*(z_t)) \leq \sum_{t=1}^{T}\frac{\gamma}{\beta_t}\sum_{s=1}^{t-1}\beta_s(f_t(z_s) - f_*(z_s))^2 + \gamma\lambda\sum_{t=1}^{T}\frac{1}{\beta_t} + \left(1 + \frac{1}{4\gamma}\right)\dim_{\epsilon,T}(\mathcal{F}).$$

# B  Proof of Main Theorem

In this section, we first provide a more general version of Theorem 5.4:

**Theorem B.1.** Define

$$Z_t = -\mathbb{E}_{S_{t-1},x_t}\log\mathbb{E}_{\widetilde{f}\sim p_0}\exp\left(-\sum_{s=1}^{t-1}\bar{\sigma}_s^{-2}\Delta L(\widetilde{f}, x_s, a_s, r_s) + \frac{c\sqrt{\Lambda_t}}{\bar{\sigma}_t^2}\mathrm{FG}_t(\widetilde{f})\right) \qquad (B.1)$$

$$Z = 1 \vee \sup_{\{\bar{\sigma}_t\}_{t\in[T]}}\max_{t\in[T]}Z_t. \qquad (B.2)$$

Suppose that the parameters are set as in (4.2), and the hyperparameters are

$$\lambda = 1, \quad \epsilon = \alpha^{-2}, \quad c = 2\sqrt{Z/\mathrm{dc}_{\lambda,\epsilon,T}(\mathcal{F})}$$

Then the total regret satisfies

$$\mathbb{E}[\text{Regret}(T)] \leq \frac{9}{4}\sqrt{(\alpha^2 T + \Lambda) Z \, \text{dc}_{\lambda,\epsilon,T}(\mathcal{F})} + \text{dc}_{\lambda,\epsilon,T}(\mathcal{F}).$$

To prove Theorem B.1, we need the following lemma:

**Lemma B.2.** Under the conditions of Theorem B.1, the following inequality holds:

$$\mathbb{E}_{S_{t-1},x_t}\mathbb{E}_{\widetilde{f}\sim p_t}\left[\frac{\bar{\sigma}_t^2}{2c\sqrt{\Lambda}}\sum_{s=1}^{t-1}\bar{\sigma}_s^{-2}\,\text{LS}_s(\widetilde{f}) - \text{FG}_t(\widetilde{f})\right] \leq \frac{\bar{\sigma}_t^2}{c\sqrt{\Lambda_t}}Z_t.$$

We show the proof of Lemma B.2 in Appendix C.1. We now provide the proof of Theorem B.1:

*Proof of Theorem B.1.* The regret can be decomposed as

$$\text{Regret}(t) = \sum_{t=1}^{T}[f_*(x_t, a_t^*) - f_*(x_t, a_t)] = \sum_{t=1}^{T}[f_t(x_t, a_t) - f_*(x_t, a_t) - \text{FG}_t(f_t)]$$

$$\leq \sum_{t=1}^{T}\left[\frac{\gamma}{\beta_t}\sum_{s=1}^{t-1}\beta_s\,\text{LS}_s(f_t) - \text{FG}_t(f_t)\right] + \gamma\lambda\sum_{t=1}^{T}\frac{1}{\beta_t} + \left(1 + \frac{1}{4\gamma}\right)\text{dc}_{\lambda,\epsilon,T}(\mathcal{F}), \qquad \text{(B.3)}$$

where the equality holds due to the optimality of $a_t$, and the inequality holds due to the definition of the generalized decoupling coefficient. Taking the expectation on both sides of (B.3), we have

$$\mathbb{E}[\text{Regret}(T)] \leq \sum_{t=1}^{T}\mathbb{E}\left[\frac{\gamma}{\beta_t}\sum_{s=1}^{t-1}\beta_s\,\text{LS}_s(f_t) - \text{FG}_t(f_t)\right] + \gamma\lambda\sum_{t=1}^{T}\frac{1}{\beta_t} + \left(1 + \frac{1}{4\gamma}\right)\text{dc}_{\lambda,\epsilon,T}(\mathcal{F})$$

$$= \sum_{t=1}^{T}\mathbb{E}_{S_{t-1},x_t}\mathbb{E}_{\widetilde{f}\sim p_t}\left[\frac{\gamma}{\beta_t}\sum_{s=1}^{t-1}\beta_s\,\text{LS}_s(\widetilde{f}) - \text{FG}_t(\widetilde{f})\right] + \gamma\lambda\sum_{t=1}^{T}\frac{1}{\beta_t} + \left(1 + \frac{1}{4\gamma}\right)\text{dc}_{\lambda,\epsilon,T}(\mathcal{F}),$$

where the equality holds due to the double expectation theorem and because neither $\text{LS}_s(f_t)$ nor $\text{FG}_t(f_t)$ explicitly contain $a_t$. By selecting $\beta_t = \bar{\sigma}_t^{-2}$ and $\gamma = 1/(2c\sqrt{\Lambda_T})$, we can use Lemma B.2 to further bound the regret:

$$\mathbb{E}[\text{Regret}(T)] \leq \frac{\lambda\sqrt{\Lambda_T}}{2c} + \left(1 + \frac{c\sqrt{\Lambda_T}}{2}\right)\text{dc}_{\lambda,\epsilon,T}(\mathcal{F}) + \sum_{t=1}^{T}\frac{\bar{\sigma}_t^2}{c\sqrt{\Lambda_t}}Z_t$$

$$\leq \frac{\lambda\sqrt{\Lambda_T}}{2c} + \left(1 + \frac{c\sqrt{\Lambda_T}}{2}\right)\text{dc}_{\lambda,\epsilon,T}(\mathcal{F}) + Z\sum_{t=1}^{T}\frac{\bar{\sigma}_t^2}{c\sqrt{\Lambda_t}}$$

$$\leq \frac{\lambda\sqrt{\Lambda_T}}{2c} + \left(1 + \frac{c\sqrt{\Lambda_T}}{2}\right)\text{dc}_{\lambda,\epsilon,T}(\mathcal{F}) + 2\frac{Z\sqrt{\Lambda_T}}{c}$$

where the second inequality holds due to the definition of $Z$, and the last inequality holds due to Lemma D.2. Plugging in $c = 2\sqrt{Z/\text{dc}_{\lambda,\epsilon,T}(\mathcal{F})}$, we have

$$\mathbb{E}[\text{Regret}(T)] \leq \frac{\lambda}{4}\sqrt{\frac{\text{dc}_{\lambda,\epsilon,T}(\mathcal{F})\Lambda_T}{Z}} + 2\sqrt{Z\,\text{dc}_{\lambda,\epsilon,T}(\mathcal{F})\Lambda_T} + \text{dc}_{\lambda,\epsilon,T}(\mathcal{F})$$

$$\leq \frac{9}{4}\sqrt{Z\,\text{dc}_{\lambda,\epsilon,T}(\mathcal{F})\Lambda_T} + \text{dc}_{\lambda,\epsilon,T}(\mathcal{F})$$

$$\leq \frac{9}{4}\sqrt{(\alpha^2 T + \Lambda) Z\,\text{dc}_{\lambda,\epsilon,T}(\mathcal{F})} + \text{dc}_{\lambda,\epsilon,T}(\mathcal{F}),$$

where the second inequality holds because $\lambda = 1$ and $Z \geq 1$, and the last inequality holds because $\Lambda_T = \sum_{t=1}^{T}\max\{\sigma_t^2, \alpha^2\} \leq \sum_{t=1}^{T}(\sigma_t^2 + \alpha^2)$. $\qquad\square$

In order to prove Theorem 5.4 from Theorem B.1, we note that $Z \leq \log|\mathcal{F}|$ using the argument in Section 6. By selecting $\alpha = 1/\sqrt{T}$, we can prove Theorem B.1, and by selecting $\alpha$ to be a even smaller number, we can prove further shave the term $\alpha^2 T + \Lambda$.

In order to deal with the infinite function class, we need the following lemma to characterize $Z$:

**Lemma B.3.** Suppose that $\sigma_t$ is uniformly bounded by $\sigma$, and the hyperparameter $\alpha$ satisfies $\alpha \leq 1$. Define

$$\delta = \frac{\alpha^2}{2T}\min\{1, \sigma^{-1}\}.$$

Then $Z$ satisfies

$$Z \leq \log \frac{1}{p_0(\mathcal{F}_\delta(f_*))} + \frac{\alpha}{2} + \frac{c}{\sqrt{T}}.$$

For the linear reward function class where $\Theta$ is the unit ball, we have $-\log p_0(\mathcal{F}_\delta(f_*)) = \widetilde{\mathcal{O}}(d)$ according to Zhang (2022). Therefore, the regret of FGTS-VA in linear contextual bandits is $\widetilde{\mathcal{O}}(d\sqrt{\Lambda} + d)$, even if $\mathcal{F}$ is not the $\epsilon$-net. [2]

## C Proof of Lemmas in Appendix B

### C.1 Proof of Lemma B.2

*Proof.* We aim to apply Lemma D.1, so for any $s \in [t-1]$, we define $F_s(\widetilde{f}, x_s, a_s, r_s) = \bar{\sigma}_s^{-2}\Delta L(\widetilde{f}, x_s, a_s, r_s)$, then by Lemma D.1, we have

$$0 = -\log \mathbb{E}_{\widetilde{f}\sim p_0}\mathbb{E}_{S_{t-1},x_t}\exp\left(\sum_{s=1}^{t-1}\xi_s(\widetilde{f}, x_s, a_s, r_s)\right)$$

$$\leq -\mathbb{E}_{S_{t-1},x_t}\log \mathbb{E}_{\widetilde{f}\sim p_0}\exp\left(\sum_{s=1}^{t-1}\xi_s(\widetilde{f}, x_s, a_s, r_s)\right), \tag{C.1}$$

where the inequality holds due to the Jensen's inequality. We then use Lemma 6.1 twice:

$$-\log \mathbb{E}_{\widetilde{f}\sim p_0}\exp\left(\sum_{s=1}^{t-1}\xi_s(\widetilde{f}, x_s, a_s, r_s)\right) \leq \mathbb{E}_{\widetilde{f}\sim p_t}\left[-\sum_{s=1}^{t-1}\xi_s(\widetilde{f}, x_s, a_s, r_s)\right] + \mathrm{KL}(p_t\|p_0), \tag{C.2}$$

$$-\log \mathbb{E}_{\widetilde{f}\sim p_0}\exp\left(-\sum_{s=1}^{t-1}\bar{\sigma}_s^{-2}\Delta L(\widetilde{f}, x_s, a_s, r_s) + \lambda_t\,\mathrm{FG}_t(\widetilde{f})\right)$$

$$= \mathbb{E}_{\widetilde{f}\sim p_t}\left[\sum_{s=1}^{t-1}\bar{\sigma}_s^{-2}\Delta L(\widetilde{f}, x_s, a_s, r_s) - \lambda_t\,\mathrm{FG}_t(\widetilde{f})\right] + \mathrm{KL}(p_t\|p_0). \tag{C.3}$$

Furthermore, since

$$F_s(\widetilde{f}, x_s, a_s, r_s) = \bar{\sigma}_s^{-2}[(r_s - \widetilde{f}(x_s, a_s))^2 - (r_s - f_*(x_s, a_s))^2]$$

$$= \bar{\sigma}_s^{-2}[(\epsilon_s + f_*(x_s, a_s) - \widetilde{f}(x_s, a_s))^2 - \epsilon_s^2]$$

$$= \bar{\sigma}_s^{-2}\,\mathrm{LS}_s(\widetilde{f}) - 2\epsilon_s\bar{\sigma}_s^{-2}(\widetilde{f}(x_s, a_s) - f_*(x_s, a_s)),$$

we have

$$\log \mathbb{E}[\exp(-F_s(\widetilde{f}, x_s, a_s, r_s)|\mathcal{G}_s)]$$

$$= -\bar{\sigma}_s^{-2}\Delta L_s(\widetilde{f}) + \mathbb{E}[\exp(2\epsilon_s\bar{\sigma}_s^{-2}(\widetilde{f}(x_s, a_s) - f_*(x_s, a_s)))|\mathcal{G}_t]$$

$$\leq -\bar{\sigma}_s^{-2}\,\mathrm{LS}_s(\widetilde{f}) + \sigma_s^2\bar{\sigma}_s^{-4}/2\,\mathrm{LS}_s(\widetilde{f})$$

$$\leq -\bar{\sigma}_s^{-2}/2 \cdot \mathrm{LS}_s(\widetilde{f}), \tag{C.4}$$

where the first inequality holds due to Assumption 3.1, and the second inequality holds because $\sigma_s \leq \bar{\sigma}_s$. Plugging (C.2), (C.3), and (C.4) into (C.1), we have

$$0 \leq \mathbb{E}_{S_{t-1},x_t}\left[\mathbb{E}_{\widetilde{f}\sim p_t}\left[-\sum_{s=1}^{t-1}\xi_s(\widetilde{f}, x_s, a_s, r_s)\right] + \mathrm{KL}(p_t\|p_0)\right]$$

---

[2]Although $Z_T$ has additional terms in $\alpha$ and $c$, they can be shaved by carefully choosing hypermeters in Theorem B.1.

$$= \mathbb{E}_{S_{t-1}, x_t} \Bigg[ \mathbb{E}_{\widetilde{f} \sim p_t} \Bigg[ \sum_{s=1}^{t-1} \Delta L(\widetilde{f}, x_s, a_s, r_s) - \lambda_t \, \mathrm{FG}_t(\widetilde{f}) \Bigg] + \mathrm{KL}(p_t \| p_0)$$

$$+ \sum_{s=1}^{t-1} \mathbb{E}_{\widetilde{f} \sim p_t} \log \mathbb{E}[\exp(-\Delta L(\widetilde{f}, x_s, a_s, r_s)) | \mathcal{G}_t] + \lambda_t \mathbb{E}_{\widetilde{f} \sim p_t} \mathrm{FG}_t(\widetilde{f}) \Bigg]$$

$$\leq - \mathbb{E}_{S_{t-1}, x_t} \log \mathbb{E}_{\widetilde{f} \sim p_0} \exp \left( - \sum_{s=1}^{t-1} \bar{\sigma}_s^{-2} \Delta L(\widetilde{f}, x_s, a_s, r_s) + \lambda_t \, \mathrm{FG}_t(\widetilde{f}) \right)$$

$$+ \mathbb{E}_{S_{t-1}, x_t} \mathbb{E}_{\widetilde{f} \sim p_t} \Bigg[ - \sum_{s=1}^{t-1} \frac{\bar{\sigma}_s^{-2}}{2} \mathrm{LS}_s(\widetilde{f}) + \lambda_t \, \mathrm{FG}_t(\widetilde{f}) \Bigg].$$

Rearranging terms and plugging in $\lambda_t = c\bar{\sigma}_t^{-2} \sqrt{\Lambda_t}$, we obtain

$$\mathbb{E}_{S_{t-1}, x_t} \mathbb{E}_{\widetilde{f} \sim p_t} \Bigg[ \frac{\bar{\sigma}_t^2}{2c\sqrt{\Lambda}} \sum_{s=1}^{t-1} \bar{\sigma}_s^{-2} \mathrm{LS}_s(\widetilde{f}) - \mathrm{FG}_t(\widetilde{f}) \Bigg]$$

$$\leq \mathbb{E}_{S_{t-1}, x_t} \mathbb{E}_{\widetilde{f} \sim p_t} \Bigg[ \frac{\bar{\sigma}_t^2}{2c\sqrt{\Lambda_t}} \sum_{s=1}^{t-1} \bar{\sigma}_s^{-2} \mathrm{LS}_s(\widetilde{f}) - \mathrm{FG}_t(\widetilde{f}) \Bigg]$$

$$\leq \frac{-\bar{\sigma}_t^2}{c\sqrt{\Lambda_t}} \mathbb{E}_{S_{t-1}, x_t} \log \mathbb{E}_{\widetilde{f} \sim p_0} \exp \left( - \sum_{s=1}^{t-1} \bar{\sigma}_s^{-2} \Delta L(\widetilde{f}, x_s, a_s, r_s) + \frac{c\sqrt{\Lambda_t}}{\bar{\sigma}_t^2} \, \mathrm{FG}_t(\widetilde{f}) \right).$$

where the first inequality holds because $\Lambda_t \leq \Lambda$. $\qquad \square$

## C.2 Proof of Lemma B.3

*Proof of Lemma B.3.* We note that for any $f \in \mathcal{F}_\delta(f_*)$, we have

$$- \sum_{s=1}^{t-1} \bar{\sigma}_s^{-2} \Delta L(f, x_s, a_s, r_s) + \frac{c\sqrt{\Lambda_t}}{\bar{\sigma}_t^2} \, \mathrm{FG}_t(f)$$

$$= \sum_{s=1}^{t-1} \bar{\sigma}_s^{-2} [2\epsilon_s(f(x_s, a_s) - f_*(x_s, a_s)) - \mathrm{LS}_s(f)] + \frac{c\sqrt{\Lambda_t}}{\bar{\sigma}_t^2} \, \mathrm{FG}_t(f)$$

$$\geq - \sum_{s=1}^{t-1} \bar{\sigma}_s^{-2}(|\epsilon_s|\delta + \delta^2) - \frac{c\delta\sqrt{\Lambda_t}}{\bar{\sigma}_t^2} \qquad\qquad (\text{C.5})$$

Therefore, $-Z_t$ can be lower bounded as

$$-Z_t = \mathbb{E}_{S_{t-1}, x_t} \log \mathbb{E}_{\widetilde{f} \sim p_0} \exp \left( - \sum_{s=1}^{t-1} \bar{\sigma}_s^{-2} \Delta L(\widetilde{f}, x_s, a_s, r_s) + \frac{c\sqrt{\Lambda_t}}{\bar{\sigma}_t^2} \, \mathrm{FG}_t(\widetilde{f}) \right)$$

$$\geq \mathbb{E}_{S_{t-1}, x_t} \log \left( p_0(\mathcal{F}_\delta(f_*)) \inf_{f \in \mathcal{F}_\delta} \exp \left( - \sum_{s=1}^{t-1} \bar{\sigma}_s^{-2} \Delta L(f, x_s, a_s, r_s) + \frac{c\sqrt{\Lambda_t}}{\bar{\sigma}_t^2} \, \mathrm{FG}_t(f) \right) \right)$$

$$\geq \log p_0(\mathcal{F}_\delta(f_*)) + \mathbb{E}_{S_{t-1}, x_t} \Bigg[ - \sum_{s=1}^{t-1} \bar{\sigma}_s^{-2}(|\epsilon_s|\delta + \delta^2) - \frac{c\delta\sqrt{\Lambda_t}}{\bar{\sigma}_t^2} \Bigg]$$

$$\geq \log p_0(\mathcal{F}_\delta(f_*)) - \sum_{s=1}^{t-1} \left( \frac{\delta}{2\bar{\sigma}_s} + \frac{\delta^2}{\bar{\sigma}_s^2} \right) - \frac{c\delta\sqrt{\Lambda_t}}{\bar{\sigma}_t^2},$$

where the first inequality holds because for any function $F(x) \geq 0$, we have $\mathbb{E}[F(x)] \geq \mathbb{E}[F(x) \mathbb{1}[x \in \mathcal{C}]] \geq p(\mathcal{C})\mathbb{E}[\inf_{x \in \mathcal{C}} F(x)]$, the second inequality holds due to (C.5), and the last inequality holds because $\mathbb{E}|\epsilon_s| \leq \sqrt{\mathbb{E}\epsilon_s^2} \leq \sqrt{\sigma_s^2/4} \leq \bar{\sigma}_s/2$. By choosing $\delta = \alpha^2 \min\{1/\sigma, 1\}/(2T)$, we have

$$Z_t \leq \log \frac{1}{p_0(\mathcal{F}_\delta(f_*))} + \frac{\alpha}{4} + \frac{\alpha^2}{4T} + \frac{c}{2\sqrt{T}} \leq \log \log \frac{1}{p_0(\mathcal{F}_\delta(f_*))} + \frac{\alpha}{2} + \frac{c}{\sqrt{T}},$$

where the second inequality holds because $\alpha \leq$ and $T \geq 1$. $\qquad \square$

# D   Auxiliary Lemmas

**Lemma D.1.** For any function $F_t : \mathcal{F} \times \mathcal{X} \times \mathcal{A} \times \mathbb{R} \to \mathbb{R}$, define

$$\xi_t(\widetilde{f}, x_t, a_t, r_t) = -F_t(\widetilde{f}, x_t, a_t, r_t) - \log \mathbb{E}[\exp(-F_t(\widetilde{f}, x_t, a_t, r_t))|\mathcal{G}_t].$$

Then for all $t$ and all $\widetilde{f} \in \mathcal{F}$, we have

$$\mathbb{E}_{S_t} \exp \left( \sum_{s=1}^{t} \xi_s(\widetilde{f}, x_s, a_s, r_s) \right) = 1.$$

*Proof.* We prove the lemma by induction. The property holds trivially for $t = 0$. Now suppose that the lemma holds for $t - 1$, then note that

$$\begin{aligned}
&\mathbb{E}[\exp(\xi_t(\widetilde{f}, x_t, a_t, r_t))|\mathcal{G}_t] \\
&= \mathbb{E}\left[ \exp\left( -F_t(\widetilde{f}, x_t, a_t, r_t) - \log \mathbb{E}[\exp(-F_t(\widetilde{f}, x_t, a_t, r_t))|\mathcal{G}_t] \right) \Big| \mathcal{G}_t \right] \\
&= \mathbb{E}\left[ \frac{\exp(-F_t(x_t, a_t, r_t))}{\mathbb{E}[\exp(-F_t(\widetilde{f}, x_t, a_t, r_t))|\mathcal{G}_t]} \Big| \mathcal{G}_t \right] = 1, \quad\quad\quad\quad\quad (\text{D.1})
\end{aligned}$$

where the first equality holds due to the definition of $\xi_t$, and the last equality holds because the denominator belongs to $\mathcal{G}_t$. We then have

$$\begin{aligned}
&\mathbb{E}_{S_t} \exp \left( \sum_{s=1}^{t} \xi_s(\widetilde{f}, x_s, a_s, r_s) \right) \\
&= \mathbb{E}\left[ \exp \left( \sum_{s=1}^{t-1} \xi_s(\widetilde{f}, x_s, a_s, r_s) \right) \cdot \mathbb{E}[\exp(\xi_t(\widetilde{f}, x_s, a_s, r_s))|\mathcal{G}_t] \right] \\
&= \mathbb{E}\left[ \exp \left( \sum_{s=1}^{t-1} \xi_s(\widetilde{f}, x_s, a_s, r_s) \right) \right] = 1
\end{aligned}$$

where the first equality holds due to the double expectation theorem and because $\xi_s(\widetilde{s}, x_s, a_s, r_s) \in \mathcal{G}_t$ for all $s \in [t - 1]$, the second equality holds due to (D.1), and the last equality holds due to the induction hypothesis. Combining all the above, the lemma holds for all $t$ by induction.  $\square$

**Lemma D.2.** Let $\bar{\sigma}_t$ and $\Lambda_t$ be defined in (4.2). Then the following property holds:

$$\sum_{t=1}^{T} \frac{\bar{\sigma}_t^2}{\sqrt{\Lambda_t}} \le 2\sqrt{\Lambda_T}.$$

*Proof.* We note that

$$\sqrt{\Lambda_T} = \sum_{t=1}^{T} (\sqrt{\Lambda_t} - \sqrt{\Lambda_{t-1}}) = \sum_{t=1}^{T} \frac{\Lambda_t - \Lambda_{t-1}}{\sqrt{\Lambda_t} + \sqrt{\Lambda_{t-1}}} \ge \sum_{t=1}^{T} \frac{\bar{\sigma}_t^2}{2\sqrt{\Lambda_t}},$$

where the first equality holds due to the telescope sum and because $\Lambda_0 = 0$, and the inequality holds because $\Lambda_{t-1} \le \Lambda_t$ and $\bar{\sigma}_t^2 = \Lambda_t - \Lambda_{t-1}$.  $\square$

