# OpenReview forum: "Variance-Aware Feel-Good Thompson Sampling for Contextual Bandits"
_NeurIPS.cc/2025/Conference — NeurIPS 2025 poster_

### Official Review · Reviewer_jEKf · 2025-06-25

**Clarity:** 3
**Significance:** 3
**Originality:** 3
**Rating:** 5
**Confidence:** 4

**Summary:**

This paper brings the feel-good Thompson sampling (FGTS) algorithm into the second-order (variance-aware) contextual bandit. They show that the regret of the algorithm can upper bounded by the decoupling coefficient, specifically, if the decoupling coefficient of the function class is d, then the regret of variance-aware FGTS is bounded by $\sqrt{d*\Sigma*\log F}+d$. This is the first paper to provide variance aware guarantees for feel good Thompson sampling algorithm.

**Questions:**

I have the following questions about this paper:
1. Is there an computationally efficient way for updating the distribution and performing FGTS?
2. When the variance $\sigma_t$ is revealed to the learner after making action at round $t$, or even not revealed to the learner, does the algorithm still work?
3. In reference [1], they also present an upper bound scaled with $\sqrt{A*\Sigma}$ for the inverse-gap-weighting algorithms in that paper, does a similar guarantee holds for FGTS?
4. This paper only considers the prior to be uniform over all functions, what about other priors? Can the algorithm achieves similar guarantees, or under what assumptions over the prior can the algorithm has similar guarantees?
5. Since the Thompson sampling algorithm was firstly introduced for Bayesian regret, is there tight characterization of the variance-aware FGTS in Bayesian regret setting?

[1] Jia, Zeyu, et al. "How does variance shape the regret in contextual bandits?." Advances in Neural Information Processing Systems 37 (2024): 83730-83785.

**Ethical Concerns:**

["NO or VERY MINOR ethics concerns only"]

**Final Justification:**

This paper presents good results. Given the answers by the authors to my questions, I will maintain the score of this paper as acceptance.

**Limitations:**

Yes

**Paper Formatting Concerns:**

No paper formatting concerns in this paper

**Quality:**

3

**Strengths And Weaknesses:**

Strength:
1. This paper is well-written. All assumptions required by the algorithms are made clear and easily understood in this paper.
2. The study of variance-aware FGTS is of interest to the community, as FGTS is a kind of posterior sampling algorithm, which has better guarantee in Bayesian regret setting than other frequentist regret algorithm (like UCB or inverse-gap-weighting), and also the variance-aware study also provides tighter guarantees than the common regret bound.
3. In this paper, they considered different parameter $\eta$ across different time ste. This modification is novel and not appeared in previous feel-good TS papers.

Weakness:
1. This paper does not provide numerical experiments to demonstrate the influence of total variances and regret bound. Some numerical experiments will be much helpful to illustrate the theoretical findings.
2. There is no lower bound provided. It will be better if there exists some lower bound which shows that the regret of the upper bound is tight up to some factors.
3. The decoupling coefficient is a little bit difficult to understand than the normally used eluder dimension. It will be better if there is more explanation on how the decoupling coefficient can be bounded for some specific function classes.

---

> ### Author Rebuttal · Authors · 2025-07-31
>
> Thank you very much for the strong recommendation!
>
> ---
>
> **Q1.** This paper does not provide numerical experiments to demonstrate the influence of total variances and regret bound. Some numerical experiments will be much helpful to illustrate the theoretical findings.
>
> **A1.** We appreciate the reviewer for the suggestion. We will add experiments on the following setting:
>
> - Linear reward function: The context set has a single element, and will be omitted. The action set is $\{\pm 1/\sqrt d\}^d$. We assume that $f_*(\mathbf a)=\langle\theta_*, \mathbf a\rangle$ where $\theta_*\in B(0, 1)$.
> - Variances: We assume that the noise $\epsilon_t$ is Gaussian with variance $\sigma_t^2$ either sampled from some distribution (e.g., $\chi^2$ distribution) or being sparse (only nonzero in a few steps).
> - Baselines: We will compare `FGTS-VA`, standard `FGTS`, `SAVE`, and Weighted-OFUL+.
>
> ---
>
> **Q2.** There is no lower bound provided. It will be better if there exists some lower bound which shows that the regret of the upper bound is tight up to some factors.
>
> **A2.** We appreciate the reviewer for the suggestion. Under the assumption of finite action set, lower bounds have been shown by Jia et al. (2024). We will add the discussion of their results in the revision.
>
> ---
>
> **Q3.** The decoupling coefficient is a little bit difficult to understand than the normally used Eluder dimension. It will be better if there is more explanation on how the decoupling coefficient can be bounded for some specific function classes.
>
> **A3.** The generalized decoupling coefficient has similar intuitions of the Eluder dimension. The Eluder dimension characterizes the function class in the following way: If we know the values of the function on a historical sequence, how predictable is the value of the function on a new point? The generalized decoupling coefficient works in a similar way. The term $(f_t(z_s)-f_*(z_s))^2$ on the RHS of the definition is the error of the function on a historical sequence, and the LHS $f_t(z_t)-f_*(z_t)$ measures the error of a new point. The coefficients $\beta_t$ enables the variance-dependent bound. We have shown that the generalized decoupling coefficient is upper bounded by the generalized Eluder dimension, and is $\tilde O(d)$ for the linear reward function class.
>
> ---
>
> **Q4.** Is there an computationally efficient way for updating the distribution and performing FGTS?
>
> **A4.** The sampling can be efficiently implemented using SGLD: Suppose that the reward function class is a parameterized function class $\lbrace f_\theta: \theta\in\Theta\rbrace$. Let $q$ be the logarithm of the posterior distribution. For the sampling in step $t$, we can start from the sampled reward function in the previous step and perform the update $\theta_{t, k+1}=\theta_{t, k}+\eta\nabla_\theta q(f_{\theta_{t, k}})+\sqrt{2\eta\epsilon_{t, k}}$, where $\eta$ is a step size, and $\epsilon_{t, k}$ is a standard Gaussian random variable.
>
> ---
>
> **Q5.** When the variance $\sigma_t$ is revealed to the learner after making action at round $t$, or even not revealed to the learner, does the algorithm still work?
>
> **A5.** It would be hard to design a FGTS algorithm without knowing the variances using the current techniques: The term corresponding to step $t$ in the definition of the generalized decoupling coefficient is $\sum_{s=1}^{t-1}\frac{\beta_s}{\beta_t}(f_t(z_s)-f_*(z_s))^2$, which depends on not only $\beta_s$ for $s<t$ but also $\beta_t$ that scales with the variance of the current step. Therefore, when choosing hyperparameters given the result in (6.7), the parameters in step $t$ unavoidably depends on $\sigma_t$. Given the success of UCB-based algorithms like SAVE, we conjecture that a FGTS-type algorithm without knowing the variances can be designed if a new analysis framework is developed that (i) avoids the decoupling coefficient and (ii) derives a high-probability bound of the term $\mathrm{LS}_t$ instead of the bound in expectation.
>
> ---
>
> **Q6.** In Jia et al. (2024), they also present an upper bound scaled with $\sqrt{A\cdot\Lambda}$ for the inverse-gap-weighting algorithms in that paper, does a similar guarantee holds for FGTS?
>
> **A6.** As our algorithm is tailored for contextual bandits and function approximation, it would be hard to derive such an upper bound without incuring the size of the context set:
>
> - $\log|\mathcal F|$ is $\tilde O(|\mathcal X|\cdot|\mathcal A|)$ if we apply quantization to the output of the reward function.
> - Following the proof in Dan et al. (2021), the generalized decoupling coefficient is also $\tilde O(|\mathcal X|\cdot|\mathcal A|)$.
>
> Combining the two terms, the regret bound scales with $|\mathcal X|\cdot|\mathcal A|\cdot\sqrt{1+\Lambda}$ (ignoring logarithmic terms) when the action set and the context set are both finite.
>
> ---
>
> **Q6.** This paper only considers the prior to be uniform over all functions, what about other priors? Can the algorithm achieves similar guarantees, or under what assumptions over the prior can the algorithm has similar guarantees?
>
> **A6.** The assumption of finite function class in Theorem 5.4 is only for the clarity of presentation. In the appendix, we present Theorem B.1 that does not depend on the assumption of finite function class. Combined with Lemma B.3, we can see that the regret bound scales with $\sqrt{Z}$, where the quantity $Z$ is related to the measure induced by the prior distribution on the neighborhood of the ground truth. Therefore, `FGTS-VA` should perform well as long as the prior distribution has adequate coverage around the ground truth.
>
> ---
>
> **Q7.** Since the Thompson sampling algorithm was firstly introduced for Bayesian regret, is there tight characterization of the variance-aware FGTS in Bayesian regret setting?
>
> **A7.** Since the Bayesian regret is the average of the frequentist regret studied in this paper, the regret bound of `FGTS-VA` applies naturally to Bayesian regret.
>
> ---
>
> Jia et al. How Does Variance Shape the Regret in Contextual Bandits? 2024.

---

> > ### Comment · Reviewer_jEKf · 2025-08-02
> >
> > Thank you to the authors for their response. I’m still curious about the SGLD algorithm you mentioned. Is there any theoretical guarantee that the sampling distribution converges to the posterior distribution? And if so, what is the rate of convergence?

---

> ### Author Response · Authors · 2025-08-02
>
> Thank you very much for the great question! There have been numerous works studying how fast the distribution of SGLD iterates converge to the underlying ground truth distribution $p_*$. A important result in Dadalyan and Karagulyan (2017, Theorem 4) shows the convergence rate under the assumption that the logarithm of the $\log p_*$ is $M$-smooth and $m$-strongly concave, and a constant step size is used in a total of $K$ steps. The result measures the Wasserstein-2 distance of the the distribution in step $K$ and $p_*$, which consists of two parts: (i) the bias error that decays exponentially in $K$, and (ii) a variance term that does not blow up even when $K$ is large. Zou et al. (2021) obtained the convergence bound on KL divergence, but with relaxed assumptions.
>
> Dadalyan and Karagulyan, User-friendly guarantees for the Langevin Monte Carlo with inaccurate gradient. 2017.
>
> Zou et al. Faster Convergence of Stochastic Gradient Langevin Dynamics for Non-Log-Concave Sampling. 2021.

---

> > ### Comment · Reviewer_jEKf · 2025-08-05
> >
> > Thank you for the clarification.

---

### Official Review · Reviewer_5NwX · 2025-07-03

**Clarity:** 3
**Significance:** 2
**Originality:** 3
**Rating:** 4
**Confidence:** 3

**Summary:**

The paper introduces **FGTS-VA**, a variance-aware variant of Feel-Good Thompson Sampling for contextual bandits.  The algorithm samples a reward model from a modified posterior that (i) weights squared-loss terms by the per-round noise variance and (ii) adds a “feel-good” exploration bonus.  A new **generalized decoupling coefficient** \(dc_{\lambda,\varepsilon,T}\) is defined to control the dependence between the sampled model and the action it optimizes; it upper-bounds to \(O(d)\) for linear models and to the generalized Eluder dimension for arbitrary function classes.  The authors prove an expected regret which matches the best known UCB-style bounds for linear bandits and collapses to \(O(\tilde d)\) in deterministic settings.

**Questions:**

### Questions and Suggestions
1. **Variance-revealing assumption** – Many real systems do not expose \(\sigma_t^2\).  Could the method be adapted with online variance estimation, or maybe a doubling scheme?  Even a heuristic discussion would help.
2. **Computability of \(dc_{\lambda,\varepsilon,T}\)** – In practice, how would one estimate or bound the generalized decoupling coefficient for non-linear models?  Any thoughts or heuristics?
3. **Finite vs. infinite \(\mathcal F\)** – The main theorem requires a finite function class with a uniform prior. Remark 5.5 talks about the linear case with F being set to an \epsilon net of the unit ball. What about general function classes? A complete analysis with an epsilon net argument is missing. Also in comparison to the IGW methods (https://arxiv.org/pdf/2107.02237, https://arxiv.org/pdf/2107.02237), if we assume finite arms K, would the bound only depend on K and not something like log |F| or Eluder dimension?
4. **Empirical check** – A small simulation (e.g., linear bandit with heteroscedastic noise) comparing FGTS-VA to LinVDTS and weighted OFUL would demonstrate that the theory transfers to practice.

**Ethical Concerns:**

["NO or VERY MINOR ethics concerns only"]

**Final Justification:**

The submission is a good theoretical contribution, and as such I support it's acceptance

**Limitations:**

yes

**Quality:**

3

**Strengths And Weaknesses:**

### Strengths and Weaknesses
**Quality** – The theoretical analysis is careful: bounds hold for finite function classes. Matching optimal variance-dependent regret for linear bandits closes a known gap between UCB and Thompson-sampling methods.  However, the work is entirely theoretical; there is no empirical demonstration.

**Clarity** – Algorithm 1 and Table 1 are easy to follow, and key assumptions are explicit.  The proofs are long and notation-heavy; readers unfamiliar with prior FGTS analyses may struggle.

**Significance** – Providing the first variance-aware, horizon-free FGTS algorithm is a meaningful advance for bandit theory.  Practical impact is limited because the algorithm assumes each round’s variance \(\sigma_t^2\) is revealed. This is in contrast to other existing variance aware methods.

**Originality** – Extending FGTS with variance weighting and introducing the generalized decoupling coefficient are novel ideas, though they build on established FGTS and UCB techniques.

---

> ### Author Rebuttal · Authors · 2025-07-31
>
> Thank you very much for the positive review!
>
> **Q1.** The work is entirely theoretical; there is no empirical demonstration.
>
> A small simulation (e.g., linear bandit with heteroscedastic noise) comparing FGTS-VA to LinVDTS and weighted OFUL would demonstrate that the theory transfers to practice.
>
> **A1.** We appreciate the reviewer for the suggestion.  We will add experiments on the following setting:
>
> - Linear reward function: The context set has a single element, and will be omitted. The action set is $\{\pm 1/\sqrt d\}^d$. We assume that $f_*(\mathbf a)=\langle\theta_*, \mathbf a\rangle$ where $\theta_*\in B(0, 1)$.
> - Variances: We assume that the noise $\epsilon_t$ is Gaussian with variance $\sigma_t^2$ either sampled from some distribution (e.g., $\chi^2$ distribution) or being sparse (only nonzero in a few steps).
> - Baselines: We will compare `FGTS-VA`, standard `FGTS`, `SAVE`, and Weighted-OFUL+.
>
> ---
>
> **Q2.** The proofs are long and notation-heavy; readers unfamiliar with prior FGTS analyses may struggle.
>
> **A2.** We have provided Section 6 as an overview of proof techniques. We will also add more details about how the decoupling coefficient is used, which is a crucial technical tool for FGTS-based algorithms.
>
> ---
>
> **Q3.** Practical impact is limited because the algorithm assumes each round’s variance is revealed. This is in contrast to other existing variance aware methods.
>
> Many real systems do not expose $\sigma_t^2$. Could the method be adapted with online variance estimation, or maybe a doubling scheme? Even a heuristic discussion would help.
>
> **A3.** We believe that the setting where variances are known is a starting point for developing truly variance-dependent algorithms without knowing $\sigma_t^2$. We will continue working on this direction.
>
> It would be hard to design a FGTS-based algorithm without knowing the variances using the current techniques: The term corresponding to step $t$ in the definition of the generalized decoupling coefficient is $\sum_{s=1}^{t-1}\frac{\beta_s}{\beta_t}(f_t(z_s)-f_*(z_s))^2$, which depends on not only $\beta_s$ for $s<t$ but also $\beta_t$ that scales with the variance of the current step. Therefore, when choosing hyperparameters given the result in (6.7), the parameters in step $t$ unavoidably depends on $\sigma_t$. Given the success of UCB-based algorithms like SAVE, we conjecture that a FGTS-type algorithm without knowing the variances can be designed if a new analysis framework is developed that (i) avoids the decoupling coefficient and (ii) derives a high-probability bound of the term $\mathrm{LS}_t$ instead of the bound in expectation. With this analysis framework, a doubling scheme or online variance estimates will be helpful in removing the dependence on $\sigma_t^2$.
>
> ---
>
> **Q4.** Extending FGTS with variance weighting and introducing the generalized decoupling coefficient are novel ideas, though they build on established FGTS and UCB techniques.
>
> **A4.** Beyond the generalized decoupling coefficient, we would also like to highlight our novel techniques in choosing the parameters in the posterior distribution. In UCB-based algorithms, there is no parameter $\lambda_t$ that controls feel-good exploration, and in FGTS for standard contextual bandits, $\lambda_t$ is $1/\sqrt T$. We are the first to use the partial sum of the variances $\Lambda_t$ in variance-aware bandit algorithms. In this way, the requirement of the total sum of variances $\Lambda$ can be avoided.
>
> ---
>
> **Q5.** In practice, how would one estimate or bound the generalized decoupling coefficient for non-linear models? Any thoughts or heuristics?
>
> **A5.** There has been no work studying the Eluder dimension for practical neural networks. We hereby provide a heuristic of estimating the generalized decoupling coefficient for MLP: We can modify the analysis in Appendix A.1. We can define $\phi_t$ to be the output of the last hidden layer. However, since the network weights change across time, we need to characterize the difference between $\phi_t(z_s)$ and $\phi_s(z_s)$ using the Lipschitzness of the network.
>
> ---
>
> **Q6.** The main theorem requires a finite function class with a uniform prior. Remark 5.5 talks about the linear case with F being set to an $\varepsilon$-net of the unit ball. What about general function classes? A complete analysis with an epsilon net argument is missing.
>
> **A6.** The assumption of finite function class in Theorem 5.4 is only for the clarity of presentation, and we consider the $\varepsilon$-net for the linear function class to show that the regret bound of `FGTS-VA` recovers those of UCB-based algorithms. In the appendix, we present Theorem B.1 that does not depend on the assumption of finite function class, so the discussion about the $\varepsilon$-net for the general function class is unnecessary.
>
> ---
>
> **Q7.** Also in comparison to the IGW methods, if we assume finite arms $K$, would the bound only depend on $K$ and not something like $\log|\mathcal F|$ or the generalized decoupling coefficient?
>
> **A7.** As our algorithm is tailored for contextual bandits and function approximation, it would be hard to remove the dependence on $\log|\mathcal F|$ and the generalized decoupling coefficient without incuring the cardinality of the context set.
>
> - $\log|\mathcal F|$ is $\tilde O(|\mathcal X|\cdot|\mathcal A|)$ if we apply quantization to the output of the reward function.
> - Following the proof in Dan et al. (2021), the generalized decoupling coefficient is also $\tilde O(|\mathcal X|\cdot|\mathcal A|)$.
>
> Combining the two terms, the regret bound scales with $|\mathcal X|\cdot|\mathcal A|$ (ignoring logarithmic terms) when the action set and the context set are both finite.
>
> ---
>
> Dann et al. A provably efficient model-free posterior sampling method for episodic reinforcement learning. 2021

---

> > ### Comment · Reviewer_5NwX · 2025-08-06
> >
> > I thank the authors for the replies. My questions have been adressed.
> > However, since no empirical evidence has been provided yet, I maintain my initial evaluation "there is no empirical demonstration."
> >
> > However, I do understand that this is indeed a good theoritical contribution, and therefore support its acceptance.

---

### Official Review · Reviewer_3xSL · 2025-07-03

**Clarity:** 3
**Significance:** 2
**Originality:** 2
**Rating:** 4
**Confidence:** 3

**Summary:**

This paper proposes FGTS-VA, a new variance-aware Thompson Sampling algorithm for contextual bandits that achieves optimal regret bounds. The key innovation is incorporating variance-dependent weights and a "feel-good" exploration term into the posterior distribution, allowing it to handle general reward functions while matching the performance of UCB-based methods. Theoretical analysis introduces a generalized decoupling coefficient to prove regret bounds for linear bandits, which is minimax optimal and improves upon prior TS approaches. Though requiring known variances, this work represents significant progress in making Thompson Sampling both theoretically sound and practically competitive for variance-sensitive bandit problems.

**Questions:**

1. Is it possible to design a feel-good Thompson Sampling type algorithm without needing to know the variances?

2. Could you explain in more intuitive terms how the generalized decoupling coefficient differs from standard decoupling coefficients?

3. How does the feel-good exploration term in Type B posterior differ fundamentally from Type A in handling variance awareness? Why does Type A fail in this setting?

**Ethical Concerns:**

["NO or VERY MINOR ethics concerns only"]

**Limitations:**

yes

**Quality:**

3

**Strengths And Weaknesses:**

Strengths:

1. Theoretical guarantees of this paper are sound, when the variances are known, the upper bound of the algorithm matches the UCB-type algorithm.

2. The writing of this paper is clear and complete, and they also mentioned the technical challenges in the proofs.

Weaknesses:

1. This algorithm requires knowing the variances, but for UCB-type algorithms in the field of Variance-Aware contextual bandits, there are already good algorithms like Save+, which do not need to know the variances for prior knowledge. This means that the algorithm proposed in this paper can not match the state-of-the-art in UCB-type algorithms.

2. The algorithm design seems to be just a trivial combination of weighted linear regression and feel-good Thompson Sampling.

3. There are no empirical studies in this paper.

---

> ### Author Rebuttal · Authors · 2025-07-31
>
> Thank you very much for the positive review!
>
> ---
>
> **Q1.** This algorithm requires knowing the variances, but for UCB-type algorithms in the field of Variance-Aware contextual bandits, there are already good algorithms like Save+, which do not need to know the variances for prior knowledge. This means that the algorithm proposed in this paper can not match the state-of-the-art in UCB-type algorithms.
>
> **A1.** We believe that the setting where variances are known is a starting point for developing truly variance-dependent algorithms. We will continue working on this direction.
>
> ---
>
> **Q2.** The algorithm design seems to be just a trivial combination of weighted linear regression and feel-good Thompson Sampling.
>
> **A2.** We would like to point out a misunderstanding of the reviewer: Although the algorithm looks like FGTS + weighted linear regression, the selection of parameters $\lambda_t$ and the definition of generalized decoupling coefficient constitute novel technical contribution of our paper. In UCB-based algorithms, there is no parameter $\lambda_t$ that controls feel-good exploration, and in FGTS for standard contextual bandits, $\lambda_t$ is $1/\sqrt T$. We are the first to use the partial sum of the variances $\Lambda_t$ in variance-aware bandit algorithms. The definition of the generalized decoupling coefficient is also nontrivially different from the generalized Eluder dimension due to the introduction of parameters $\beta_t$.
>
> ---
>
> **Q3.** There are no empirical studies in this paper.
>
> **A3.** We appreciate the reviewer for the suggestion. We will add experiments on the following setting:
>
> - Linear reward function: The context set has a single element, and will be omitted. The action set is $\{\pm 1/\sqrt d\}^d$. We assume that $f_*(\mathbf a)=\langle\theta_*, \mathbf a\rangle$ where $\theta_*\in B(0, 1)$.
> - Variances: We assume that the noise $\epsilon_t$ is Gaussian with variance $\sigma_t^2$ either sampled from some distribution (e.g., $\chi^2$ distribution) or being sparse (only nonzero in a few steps).
> - Baselines: We will compare `FGTS-VA`, standard `FGTS`, `SAVE`, and Weighted-OFUL+.
>
> ---
>
> **Q4.** Is it possible to design a feel-good Thompson Sampling type algorithm without needing to know the variances?
>
> **A4.** It would be hard to design a FGTS algorithm without knowing the variances using the current techniques, mainly due to the generalized decoupling coefficient: The term corresponding to step $t$ in the definition of the generalized decoupling coefficient is $\sum_{s=1}^{t-1}\frac{\beta_s}{\beta_t}(f_t(z_s)-f_*(z_s))^2$, which depends on not only $\beta_s$ for $s<t$ but also $\beta_t$ that scales with the variance of the current step. Therefore, when choosing hyperparameters given the result in (6.7), the parameters in step $t$ unavoidably depends on $\sigma_t$. Given the success of UCB-based algorithms like SAVE, we conjecture that a FGTS-type algorithm without knowing the variances can be designed if a new analysis framework is developed that (i) avoids the decoupling coefficient and (ii) derives a high-probability bound of the term $\mathrm{LS}_t$ instead of the bound in expectation.
>
> ---
>
> **Q5.** Could you explain in more intuitive terms how the generalized decoupling coefficient differs from standard decoupling coefficients?
>
> **A5.** The term of major importance in the definition of the generalized decoupling coefficient is $\sum_{t=1}^T\sum_{s=1}^{t-1}\frac{\beta_s}{\beta_t}(f_t(z_s)-f_*(z_s))^2$. The main difference from the standard decoupling coefficient is the introduction of $\beta_t$. Intuitively, these weights makes the terms $\beta_s[(r_s-f_t(z_s))^2-(r_s-f_*(z_s))^2]$ more balanced. The additional $\frac1{\beta_t}$ serves as a normalization factor that cancels out with $\beta_s$ when the variances are homogeneous.
>
> ---
>
> **Q6.** How does the feel-good exploration term in Type B posterior differ fundamentally from Type A in handling variance awareness? Why does Type A fail in this setting?
>
> **A6.** The failure of Type A posterior attributes mainly to the analysis technique that depends on the difference of the potential $Z_t-Z_{t-1}$. The log-sum-exp term causes minor gain when the variance is small compared to when the variance is large. A major difference in Type B posterior is that it can incorporate information of the current step. It would be interesting to draw intuitions about how this difference results in the failure of Type A but success of Type B, but currently, our preference for Type B is solely due to technical considerations.

---

> > ### Comment · Reviewer_3xSL · 2025-08-04
> > **Thanks for the rebuttal**
> >
> > Thanks for the rebuttal. Your clarifications on the algorithm design and variance dependence are helpful. I appreciate the plan to include experiments and acknowledge the novelty of the generalized decoupling coefficient. I will maintain my positive rating.

---

> > > ### Author Response · Authors · 2025-08-05
> > >
> > > Many thanks for your reply and the positive rating! We would be happy to further resolve your questions and make improvements to our manuscript.

---

### Official Review · Reviewer_tepv · 2025-07-08

**Clarity:** 2
**Significance:** 3
**Originality:** 3
**Rating:** 4
**Confidence:** 4

**Summary:**

The paper studies the problem of regret minimization for contextual bandits. More specifically, they study the setting of "weak adversary with variance revealing" in which the noise (or an upper bound of the noise) is revealed to the learner at each time step. In that setting, previous work obtain a regret bound scaling with the cumulative variances $\lambda$ and the dimension $d$ at a rate $d \sqrt{\Lambda} + d$ for UCB-based methods and at a rate $d^{1.5} \sqrt{\Lambda} + d^{1.5}$ for Thompson Sampling based methods. The authors bridge this gap and propose an adaptation of the Feel-Good Thompson Sampling algorithm that attains the correct scaling with respect to the cumulative variance and the dimension. To do that, they propose a new version of the decoupling coefficient, the generalized decoupling coefficient that controls the Bellman Error in a variance-aware way. Then, they propose a new analysis of the Feel-Good Thompson Sampling algorithm that doesn't rely on the usual telescoping analysis of exponential weights. This allows them to use time-dependent learning rates and control the regret of their algorithm in a variance-aware way.

**Questions:**

- The fact that this analysis doesn't use the telescoping argument is quite surprising to me, in particular, it's not required that the learning rates are increasing or decreasing which is often the case for this type of analysis. While I agree with the technical details I would like to have a higher level understanding of the analysis. Could you provide a very high level of this part of the analyis and in particular why the telescoping is not needed anymore ?
- If it were possible to use time-varying learning rate(and still telescope Z_t - Z_t-1) with Type-A FGTS, would it be possible to adapt to the variance ?
- Is there a way to tune the other hyperparameters of the algorithm (like c) to obtain an anytime algorithm ? If not, what are the main limitations ?
- Could the techniques of this paper be adapted to the "strong" adversary case ?

**Ethical Concerns:**

["NO or VERY MINOR ethics concerns only"]

**Final Justification:**

Given the answers of the authors, I maintain my positive view of the paper.

**Limitations:**

yes

**Paper Formatting Concerns:**

Nothing to report

**Quality:**

3

**Strengths And Weaknesses:**

The main result of the paper shows that Thompson-Sampling based method can obtain the same variance-aware rate as UCB based methods. Moreover, as noted by the authors, their algorithm can accomodate time variying learning rates and doesn't require previous knowledge of the total variance. This results holds with a high degree of generality as it mostly requires bounded generalized eluder dimension.
The analysis is also insightful, they show the limitations of the usual telescoping analysis of the exponential weights algorithms when dealing with the concurrent choice of the learning rates. They show how to circumvent this limitation by avoiding the telescoping all together. The work of relating the instantaneous regret to the potential of the exponential weight is then handled by the generalized decoupling coefficient and some careful analysis. The arguments they use are (to my knowledge) new and could be used beyond the scope of this paper to analyze exponential weights algorithms.

The main limitation of the result is that the algorithms requires the knowledge of $\sigma_t$ unlike the SAVE algorithm mentionned in previous work. Also, while the algorithm is able to scale with the cumulative variance $\Lambda_t$, the algorithm is not anytime as only $\nu_t $ and $\lambda_t$ are scaling with time. The rest of the hyperparameters need to be set with knowledge of the horizon.


The analysis is somewhat difficult to follow in its current form. A significant portion of the main paper focuses on illustrating the failure modes of Type A FGTS, which results in many crucial elements of the new analysis being deferred to the appendix. As a consequence, the presentation feels scattered. It would greatly improve clarity to include a few introductory lines in the main text that summarize the gist of the analysis starting from the regret and Bellman Error Feel-Good decomposition, introducing the generalized decoupling coefficient, and then outlining the bounds on the LS and FG terms. For example, Line 307 is difficult to interpret without such context.

Additionally, there are some missing details, unclear references, and minor errors or typos that hinder comprehension:

* In the main text, pointers to the appendix are not always clear. For instance, Line 301 states: “the second inequality holds due to Assumption 3.1 with an argument similar to (6.3). (6.7) is thus completely free of the expectation of exponential terms,” but does not clearly indicate where in the appendix the relevant derivation can be found.
* Conversely, the appendix sometimes refers imprecisely to the main text. Line 461, for example, states: “we note that $Z \leq \log |\mathcal{F}|$ using the argument in Section 6,” without a clear pointer to the specific argument.
* In Line 121, there is a typo, the cumulative variance is not appearing in the regret bound.
* Line 209: The decoupling coefficient should be defined such that the stated relationship holds for any $\gamma$.
* Line 270: The function to which Jensen’s inequality is applied is not explicitly stated. Since the function involved is the log-sum-exp, whose convexity is less immediate than that of more common functions, it would be helpful to mention this explicitly—or alternatively, use Hölder’s inequality.
* Lemma 6.1 is cited as originating from Zhang (2023), but it is actually the well-known Donsker–Varadhan duality formula for the KL divergence.
* Lines 192–193 mention that $\sigma_t$ is revealed to the agent at the beginning of step $t$, and that this defines the “weak adversary” setting. This should ideally be clarified earlier in the paper, when the setting is first introduced.
* Line 216 refers to an additional term in the generalized decoupling coefficient as arising from potentially large values of $\beta_t$, with a promise of further explanation in Appendix A. However, the appendix contains little elaboration on this point.
* The choice to set the hyperparameter $\epsilon = T$ is surprising and counterintuitive, as one typically expects $\epsilon$ to be a small quantity. This can be confusing at first (I initially assumed it was a typo). Moreover, this choice introduces an additional $\log T$ factor in the generalized decoupling coefficient for the linear function class. When combined with the $\log |\mathcal{F}|$ term in the regret, it appears that the logarithmic term lies outside the square root, unlike in UCB-based methods where it is typically inside.

---

> ### Author Rebuttal · Authors · 2025-07-31
>
> **Q1 (Hyperparameters).** While the algorithm is able to scale with the cumulative variance $\Lambda_t$, the algorithm is not anytime as only $\lambda_t$ and $\mu_t$ are scaling with time. The rest of the hyperparameters need to be set with knowledge of the horizon.
>
> The choice to set the hyperparameter $\epsilon$ is surprising and counterintuitive, as one typically expects $\epsilon$ to be a small quantity. This can be confusing at first (I initially assumed it was a typo). Moreover, this choice introduces an additional $\log T$ factor in the generalized decoupling coefficient for the linear function class. When combined with the $\log|\mathcal F|$ term in the regret, it appears that the logarithmic term lies outside the square root, unlike in UCB-based methods where it is typically inside.
>
> Is there a way to tune the other hyperparameters of the algorithm (like c) to obtain an anytime algorithm ? If not, what are the main limitations ?
>
> **A1.** We would like to remark that the choice of hyperparameters $\lambda$, $\alpha$, and $\epsilon$ is less restricted than $c$, $\mu_t$, and $\lambda_t$. We first briefly discuss the reason we set the hyperparameters as in Theorem 5.4:
>
> - *Selection of $\lambda$.* We set $\lambda=1$ for simplification.
> - *Selection of $\alpha$.* The upper bound depends on $\Lambda_T=\sum_{t=1}^T\max\lbrace\sigma_t^2, \alpha^2\rbrace\le\alpha^2T+\Lambda$. We select $\alpha=1/\sqrt T$ to make $\alpha^2T=O(1)$.
> - *Selection of $\epsilon$.* $\epsilon$ is the upper bound of $\beta_t$. Since $\beta_t=\bar\sigma_t^{-2}=\sigma_t^{-2}\wedge\alpha^{-2}\le\alpha^{-2}$, we set $\epsilon=\alpha^{-2}$. We agree that $\epsilon=T$ may cause confusion because $\epsilon$ is usually small, and will change $\epsilon$ to $B$.
>
> It is possible to choose these hyperparameters to achieve an anytime algorithm, under mild assumptions. If $T$ is not known, we can still choose $\alpha=1/\sqrt{\hat T}$ if $\hat T$ is an upper bound of $T$ that can be polynomial in $T$, as long as the generalized decoupling coefficient depends logarithmically on $\epsilon$. We would disagree with the reviewer on the $\log T$ term instead of $\sqrt{\log T}$ because $\log|\mathcal F|$ does not depend on $T$.
>
> ---
>
> **Q2.** The analysis is somewhat difficult to follow in its current form. A significant portion of the main paper focuses on illustrating the failure modes of Type A FGTS, which results in many crucial elements of the new analysis being deferred to the appendix. As a consequence, the presentation feels scattered. It would greatly improve clarity to include a few introductory lines in the main text that summarize the gist of the analysis starting from the regret and Bellman Error Feel-Good decomposition, introducing the generalized decoupling coefficient, and then outlining the bounds on the LS and FG terms. For example, Line 307 is difficult to interpret without such context.
>
> **A2.** We appreciate the reviewer for the suggestion. We will add details about the application of the generalized decoupling coefficient to the analysis of the Bellman error term.
>
> ---
>
> **Q3 (Weak adversary setting).** The main limitation of the result is that the algorithms requires the knowledge of $\sigma_t$ unlike the SAVE algorithm mentioned in previous work.
>
> Could the techniques of this paper be adapted to the "strong" adversary case?
>
> Lines 192–193 mention that $\sigma_t$ is revealed to the agent at the beginning of step $t$, and that this defines the “weak adversary” setting. This should ideally be clarified earlier in the paper, when the setting is first introduced.
>
> **A3.** It would be hard to design a FGTS algorithm without knowing the variances using the current techniques: The term corresponding to step $t$ in the definition of the generalized decoupling coefficient is $\sum_{s=1}^{t-1}\frac{\beta_s}{\beta_t}(f_t(z_s)-f_*(z_s))^2$, which depends on not only $\beta_s$ for $s<t$ but also $\beta_t$ that scales with the variance of the current step. Therefore, when choosing hyperparameters given the result in (6.7), the parameters in step $t$ unavoidably depends on $\sigma_t$. Given the success of UCB-based algorithms like SAVE, we conjecture that a FGTS-type algorithm without knowing the variances can be designed if a new analysis framework is developed that (i) avoids the decoupling coefficient and (ii) derives a high-probability bound of the term $\mathrm{LS}_t$ instead of the bound in expectation.
>
> We will clarify the weak adversary setting right after Assumption 3.1 in the revision of our manuscript.
>
> ---
>
> **Q4 (Reference between main text and appendix).** In the main text, pointers to the appendix are not always clear. For instance, Line 301 states: “the second inequality holds due to Assumption 3.1 with an argument similar to (6.3). (6.7) is thus completely free of the expectation of exponential terms,” but does not clearly indicate where in the appendix the relevant derivation can be found.
>
> Conversely, the appendix sometimes refers imprecisely to the main text. Line 461, for example, states: “we note that $Z\le\log|\mathcal F|$ using the argument in Section 6,” without a clear pointer to the specific argument.
>
> Line 216 refers to an additional term in the generalized decoupling coefficient as arising from potentially large values of $\beta_t$, with a promise of further explanation in Appendix A. However, the appendix contains little elaboration on this point.
>
> **A4.** We will clarify these claims in the revision:
>
> - The derivation of (6.7) corresponds to Lines 476-477 in the appendix
> - Line 461 corresponds to (6.6) and Line 300 of the main text.
> - The case of large $\beta_t$ corresponds to the second term on the righthand side of (A.1).
>
> ---
>
> **Q5.** Type in Line 121: missing total variance; Line 270: Equality holds by Hölder’s inequality.
>
> **A5.** We appreciate the reviewer for pointing out the typo, and will continue working on refining our writings.
>
> ---
>
> **Q6.** The decoupling coefficient should be defined such that the stated relationship holds for any $\gamma$.
>
> **A6.** We would like to point out a misunderstanding of the reviewer: The requirement of $\gamma\le1$ is not a typo. The reviewer could refer to Theorem 1 of Dann et al. (2021).
>
> ---
>
> **Q7.** Lemma 6.1 is cited as originating from Zhang (2023), but it is actually the well-known Donsker–Varadhan duality formula for the KL divergence.
>
> **A7.** We appreciate the reviewer for pointing out the name of this property. We will modify the reference in the revision.
>
> ---
>
> **Q8.** The fact that this analysis doesn't use the telescoping argument is quite surprising to me, in particular, it's not required that the learning rates are increasing or decreasing which is often the case for this type of analysis. While I agree with the technical details I would like to have a higher level understanding of the analysis. Could you provide a very high level of this part of the analyis and in particular why the telescoping is not needed anymore?
>
> **A8.** The observation of the reviewer is very insightful. We provide a high-level explanation as follows:
>
> We actually implicitly use the telescope sum when bounding the generalized decoupling coefficient for the linear reward function class. The proof of Lemma A.1 (elliptical potential lemma), we can lower bound $\log\det(\Sigma_t)-\log\det(\Sigma_{t-1})$ with $1/2\cdot\beta_t\||\phi_t\||\_{\Sigma_{t-1}^{-1}}^2$ and apply telescope sum.
>
> From a high level, the regret bound should be characterized with the **information gain**, and should use the telescope sum somewhere in the proof. For Type A of FGTS, the information gain is $Z_t-Z_{t-1}$, and for Type B, the information gain is $\log\det(\Sigma_t)-\log\det(\Sigma_{t-1})$. The decoupling coefficient of type A of FGTS does not involve the information gain, and the upper bound of the decoupling coefficient for linear reward functions only uses the linear structure. Therefore, the information gain is explicit in the proof for Type A of FGTS.
>
> ---
>
> **Q9.** If it were possible to use time-varying learning rate (and still telescope $Z_t-Z_{t-1}$) with Type-A FGTS, would it be possible to adapt to the variance?
>
> **A9.** We have tried this approach, but unfortunately it does not work. We have made an explanation in Section 6.2: The telescope sum analysis relies on the RHS of (6.3) being linear in $\eta_t\mathrm{LS}_t(\tilde f)$. However, this is possible only when $\eta_t\mathrm{LS}_t(\tilde f)\lesssim1$. Since $\mathrm{LS}_t(\tilde f)$ is in the order of a constant in the worst case, $\eta_t$ cannot be $\bar\sigma_t^{-2}$ when $\sigma_t$ is close to zero.
>
> ---
>
> Dann et al. A Provably Efficient Model-Free Posterior Sampling Method for Episodic Reinforcement Learning. 2021.

---

> > ### Comment · Reviewer_tepv · 2025-08-04
> >
> > Thank you for addressing my comments, in particular, thanks for the precision on the value of lambda in question 6, I believe my reference point was the original Feel Good Thompson Sampling paper of Thong Zhang in which the decoupling coefficient is defined such that the equation holds for any $\mu>0$ but I understand that a different definition with a more restrictive $\lambda$ makes sens as well. Thanks for the clarification in Q8, so my understanding is that the learning rate does not appear in that version of the telescoping(Hence no requirement in the learning rate being monotonous), is that correct ?
> > In any case, I maintain my positive view of the paper.

---

### Decision · Program_Chairs · 2025-09-17

**Decision:**

Accept (poster)

**Comment:**

This paper contributes new algorithmic and analysis ideas to our understanding of posterior sampling-based approaches to learning contextual bandits. In particular, it proposes a novel Thompson Sampling algorithm, adapted with a 'feel-good'-style prior, that adapts to the variances of the rewards to achieve regret bounds comparable with UCB-style approaches to the problem. The theoretical analysis introduces a new notion of problem complexity by generalizing the decoupling coefficient from prior work and relating it to the classic Eluder-dimension, which may prove to be useful for further applications in sequential decision making.

On the flip side, the main concern, brought up by several referees, is the rather strong assumption of known, time-varying reward variances throughout time. The authors do clarify that this is the starting point to eventually ridding the approach of this assumption, perhaps using auxiliary estimation techniques.

Despite this concern, the referees are unanimous in their positive-leaning assessment of the paper (if not clearly recommending acceptance), and I am happy to concur with them to recommend acceptance. I urge the authors to consider the suggestions of the referees to improve the clarity of presentation in a revised version.